# Gaze patterns and brain activations in humans and marmosets in the Frith-Happé theory-of-mind animation task

Audrey Dureux[1]*, Alessandro Zanini[1], Janahan Selvanayagam[1], Ravi S Menon[1], Stefan Everling[1,2]

[1]Centre for Functional and Metabolic Mapping, Robarts Research Institute, University of Western Ontario, London, Canada; [2]Department of Physiology and Pharmacology, University of Western Ontario, London, Canada

*For correspondence:
audrey.dureux@gmail.com

Competing interest: The authors declare that no competing interests exist.

**Abstract** Theory of Mind (ToM) refers to the cognitive ability to attribute mental states to other individuals. This ability extends even to the attribution of mental states to animations featuring simple geometric shapes, such as the Frith-Happé animations in which two triangles move either purposelessly (Random condition), exhibit purely physical movement (Goal-directed condition), or move as if one triangle is reacting to the other triangle's mental states (ToM condition). While this capacity in humans has been thoroughly established, research on nonhuman primates has yielded inconsistent results. This study explored how marmosets (*Callithrix jacchus*), a highly social primate species, process Frith-Happé animations by examining gaze patterns and brain activations of marmosets and humans as they observed these animations. We revealed that both marmosets and humans exhibited longer fixations on one of the triangles in ToM animations, compared to other conditions. However, we did not observe the same pattern of longer overall fixation duration on the ToM animations in marmosets as identified in humans. Furthermore, our findings reveal that both species activated extensive and comparable brain networks when viewing ToM versus Random animations, suggesting that marmosets differentiate between these scenarios similarly to humans. While marmosets did not mimic human overall fixation patterns, their gaze behavior and neural activations indicate a distinction between ToM and non-ToM scenarios. This study expands our understanding of nonhuman primate cognitive abilities, shedding light on potential similarities and differences in ToM processing between marmosets and humans.

## Editor's evaluation

Dureux and colleagues provide important evidence regarding the capacity for mental state attribution in a highly social non-human primate species, the marmoset. Their findings suggest that marmosets and humans visually track abstract stimuli more closely during ToM animations and display differential activation of large-scale networks implicated in social processing. These findings will be of wide interest to scientists interested in social cognition.

## Introduction

Theory of Mind (ToM) refers to the capacity to ascribe mental states to other subjects (*Carruthers and Smith, 1996*; *Premack and Woodruff, 1978*). Various experimental approaches have been devised to investigate the cognitive processes involved in ToM, including tasks involving text (*Happé, 1994*), non-verbal pictures (*Sarfati et al., 1997*), false belief (*Wimmer and Perner, 1983*), and silent animations featuring geometric shapes. The latter approach is based on Heider and Simmel's observation

**eLife digest** In our daily life, we often guess what other people are thinking or intending to do, based on their actions. This ability to ascribe thoughts, intentions or feelings to others is known as Theory of Mind.

While we often use our Theory of Mind to understand other humans and interpret social interactions, we can also apply our Theory of Mind to assign feelings and thoughts to animals and even inanimate objects. For example, people watching a movie where the characters are represented by simple shapes, such as triangles, can still see a story unfold, because they infer the triangles' intentions based on what they see on the screen.

While it is clear that humans have a Theory of Mind, how the brain manages this capacity and whether other species have similar abilities remain open questions. Dureux et al. used animations showing abstract shapes engaging in social interactions and advanced brain imaging techniques to compare how humans and marmosets – a type of monkey that is very social and engages in shared childcare – interpret social cues. By comparing the eye movements and brain activity of marmosets to human responses, Dureux et al. wanted to uncover common strategies used by both species to understand social signals, and gain insight into how these strategies have evolved.

Dureux et al. found that, like humans, marmosets seem to perceive a difference between shapes interacting socially and moving randomly. Not only did their gaze linger longer on certain shapes in the social scenario, but their brain activity also mirrored that of humans viewing the same scenes. This suggests that, like humans, marmosets possess an inherent ability to interpret social scenarios, even when they are presented in an abstract form, providing a fresh perspective on primates' abilities to interpret social cues.

The findings of Dureux et al. have broad implications for our understanding of human social behavior and could lead to the development of better communication strategies, especially for individuals social cognitive conditions, such as Autism Spectrum Disorder. However, further research will be needed to understand the neural processes underpinning the interpretation of social interactions. Dureux et al.'s research indicates that the marmoset monkey may be the ideal organism to perform this research on.

that participants attribute intentional actions, human character traits, and even mental states to moving abstract shapes (*Heider and Simmel, 1944*). Subsequent studies used these animations to test the ability to ascribe mental states in autistic children (*Bowler and Thommen, 2000*; *Klin, 2000*).

In the Frith-Happé animations, a large red triangle and a small blue triangle move around the screen (*Abell et al., 2000*; *Castelli et al., 2002*; *Castelli et al., 2000*). In the Random condition, the two triangles do not interact and move purposelessly, in the Goal-Directed (GD) condition the triangles interact but in a purely physical manner (i.e. chase, dancing, fighting and leading) and in the ToM condition the two animated triangles move as if one triangle is reacting to the other's mental state (i.e. coaxing, surprising, seducing and mocking). Functional imaging studies have demonstrated that the observation of ToM compared to Random animations activates brain regions typically associated with social cognition, including dorso-medial frontal, temporoparietal, inferior and superior temporal cortical regions (*Barch et al., 2013*; *Bliksted et al., 2019*; *Castelli et al., 2000*; *Chen et al., 2023*; *Gobbini et al., 2007*; *Vandewouw et al., 2021*; *Weiss et al., 2021*; *Wheatley et al., 2007*).

Although the spontaneous attribution of mental states to moving shapes has been well established in humans, it remains uncertain whether other primate species share this capacity. There is some evidence suggesting that monkeys can attribute goals to agents with varying levels of similarity and familiarity to conspecifics, including human agents, monkey robots, moving geometric boxes, animated shapes, and simple moving dots (*Atsumi et al., 2017*; *Atsumi and Nagasaka, 2015*; *Krupenye and Hare, 2018*; *Kupferberg et al., 2013*; *Uller, 2004*). However, the findings in this area are somewhat mixed, with some studies investigating the attribution of goals to inanimate moving objects yielding inconclusive results (*Atsumi and Nagasaka, 2015*; *Kupferberg et al., 2013*). Nonhuman primates' spontaneous attribution of mental states to Frith-Happé animations is even less certain. While human subjects exhibit longer eye fixations when viewing the ToM condition compared to the Random condition of the Frith-Happé animations (*Klein et al., 2009*), a recent eye tracking

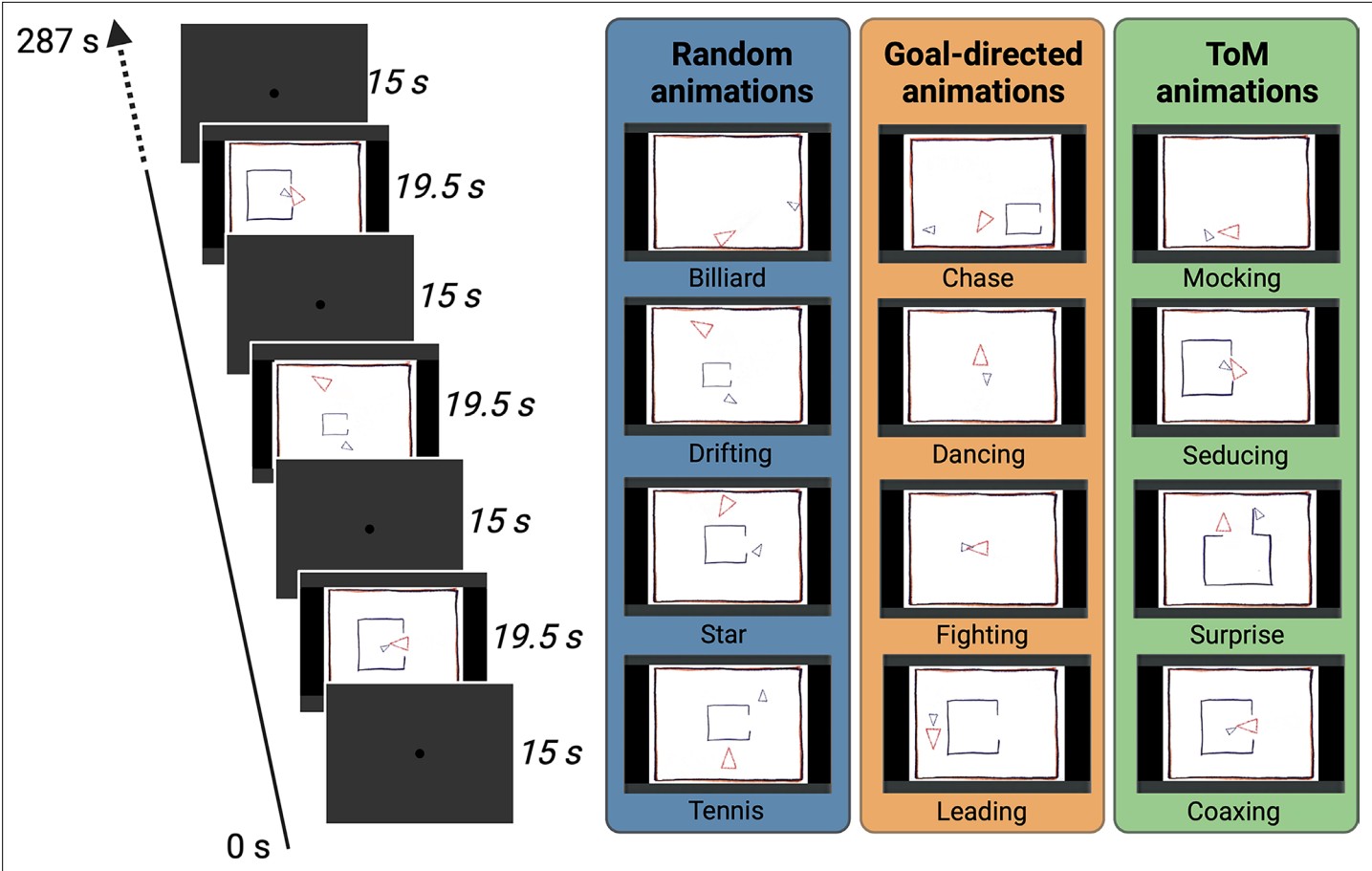

**Figure 1.** Task Design. Two different conditions of video clips resulting in eight animations were used during the scanning (ToM and Random animations), and an additional condition with four animations was used for the eye-tracking (ToM, GD and Random animations). In the ToM animations, one triangle reacted to the other triangle's mental state, whereas in the Random animations the same two triangles did not interact with each other. In the GD animations, the two triangles interact with simple intentions. Each animation video lasted 19.5 s and was separated by baseline blocks of 15 s where a central dot was displayed in the center of the screen. In the fMRI task, several runs were used with a Randomized order of the two conditions whereas in the eye-tracking task one run containing all the twelve animations once was used.

study in macaque monkeys did not observe similar differences (*Schafroth et al., 2021*). Similarly, a recent fMRI study conducted on macaques found no discernible differences in activations between ToM and random Frith-Happé animations (*Roumazeilles et al., 2021*).

In this study, we investigated the behaviour and brain activations of New World common marmoset monkeys (*Callithrix jacchus*) while they viewed Frith-Happé animations. Living in closely-knit family groups, marmosets exhibit significant social parallels with humans, including prosocial behavior, imitation, and cooperative breeding. These characteristics establish them as a promising nonhuman primate model for investigating social cognition (*Burkart et al., 2009*; *Burkart and Finkenwirth, 2015*; *Miller et al., 2016*). To directly compare humans and marmosets in their response to these animations, we employed high-speed video eye-tracking to record eye movements in eleven healthy humans and eleven marmoset monkeys. Additionally, we conducted ultra-high field fMRI scans on ten healthy humans at 7T and six common marmoset monkeys at 9.4T. These combined methods allowed us to examine the visual behavior and brain activations of both species while they observed the Frith-Happé animations.

## Results

To investigate whether marmoset monkeys, like humans, exhibit distinct processing patterns in response to the conditions in Frith-Happé animations (i.e. ToM, GD, and Random conditions), we

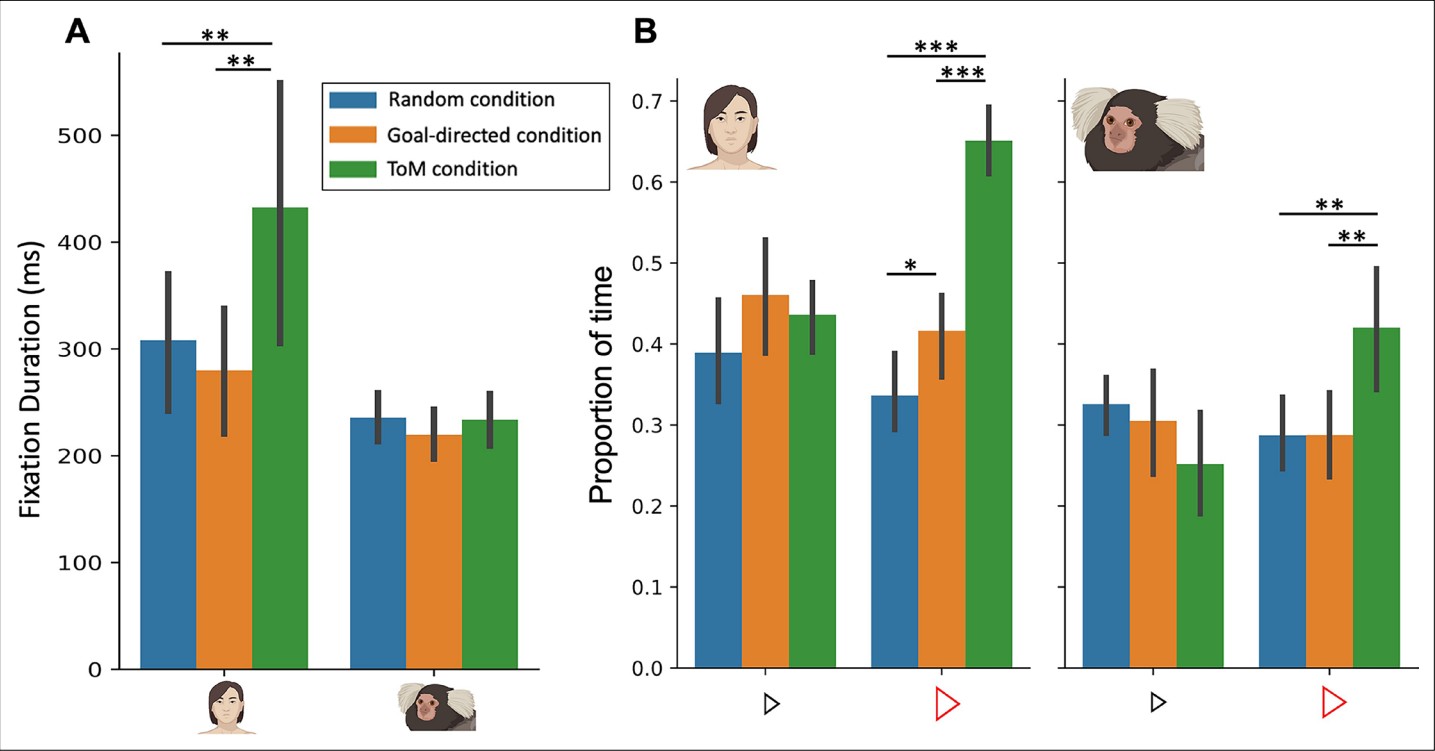

**Figure 2.** Fixation duration (A) and proportion of time looking triangles (B) in Frith-Happé's ToM, GD and Random animations in humans (left) and marmosets (right). (**A**). Bar plot depicting the fixation duration in the screen as a function of each condition. (**B**). Bar plot representing the proportion of time the radial distance between the current gaze position and each triangle was within 4 visual degrees, as a function of each condition. Green represents results obtained for ToM animation videos, orange represents results for GD animation videos and blue represents results for Random animation videos. In each graph, the left panel shows the results for 11 humans and the right panel for 11 marmosets. Each colored bar represents the group mean and the vertical bars represent the standard error from the mean. The differences between conditions were tested using ANOVA: p<0.05*, p<0.01** and p<0.001***.

compared gaze patterns and fMRI activations in both marmosets and human subjects as they watched shortened versions of the Frith-Happé animations (*Figure 1*).

## Gaze patterns for Frith-Happé's ToM, GD and Random animations in humans and marmosets

We first investigated in both humans and marmosets whether fixation durations differed between the three conditions (*Figure 2A*). By conducting mixed analyses of variance (ANOVA), with factors of species (Human vs Marmoset) and condition (ToM vs GD vs Random animation videos), we found a significant interaction between species and condition ($F_{(2,40)}$=13.9, p=<0.001, $\eta_p^2$p2.410). Here we observed longer fixation durations for ToM animation videos (*M*=432.6ms) as compared to GD videos (M=279.9ms, p=0.008) and Random videos (*M*=308.2ms, p=0.01) for humans (p=0.029) but not for marmosets (233.7ms for ToM videos, 219.6ms for GD videos and 235.6ms for Random videos, ToM vs GD: p=0.90 and ToM vs Random: p=1).This finding confirms that humans fixate longer in the ToM condition (*Klein et al., 2009*), whereas marmosets, like macaques (*Schafroth et al., 2021*), do not show this effect.

To further analyze the gaze patterns of both humans and marmosets, we next measured the proportion of time subjects looked at each of the triangles in the videos (*Figure 2B*). We conducted mixed ANOVAs on the proportion of time the radial distance between the current gaze position and each triangle was within 4 visual degrees for each triangle separately.

Importantly, we observed a significant interaction between species and condition for the proportion of time spent looking at the large red triangle ($F_{(2,40)}$=9.83, p<0.001, $\eta_p^2$p2.330). Specifically, both humans (*Figure 2B* left) and marmosets (*Figure 2B* right) spent a greater proportion of time looking at the red triangle in ToM compared to the GD and Random videos (for humans, ToM vs GD: Δ=.23,

p<0.001 and ToM vs Random: Δ=0.31, p<0.001; for marmosets, ToM vs GD: Δ=0.13, p<0.01 and ToM vs Random: Δ=0.13, p<0.01). However, while humans also allocated a greater proportion of time to the red triangle in GD compared to Random animations (Δ=0.08, p=0.05), marmosets did not show any difference between these two conditions (Δ=0.0003, p=1).

For the small blue triangle, we also observed a significant interaction of species and condition ($F_{(2,40)}$=3.54, p=0.04, $\eta_p^2$p2.151) but no significant pairwise differences were observed following Bonferroni correction. Therefore, humans and marmosets spent the same proportion of time looking at the blue triangle in the three different types of videos (for humans, ToM vs GD: Δ=-0.02, p=1, ToM vs Random: Δ=0.04, p=1 and GD vs Random: Δ=0.07, p=0.23; for marmosets, ToM vs GD: Δ=-0.05, p=0.89, ToM vs Random: Δ=0.07, p=0.66 and GD vs Random: Δ=-0.02, p=1; *Figure 2B*).

These results highlight the variation in gaze patterns observed in both humans and marmosets when their focus is directed towards the large red triangle during the viewing of ToM, GD, and Random videos. Notably, humans show a gradient of proportion of time spent looking at the red triangle across the three conditions, with the smallest proportion in Random videos and the greatest proportion in ToM videos. In contrast, marmosets exhibit a different pattern, spending more time looking at the red triangle in ToM videos, but allocating the same proportion of time to look at the red triangle in both Random and GD videos. This finding suggests that while humans demonstrate distinct attentional preferences for the red triangle across the three conditions, marmosets exhibit a similar attentional focus on the red triangle in the Random and GD conditions, but their pattern differs in the ToM condition. This suggests that marmosets process the Random and GD conditions in a similar manner, but their processing of the ToM condition is distinct, indicating a differential response to stimuli representing social interactions.

## Functional brain activations while watching ToM and Random Frith-Happé's animations in humans

Given that humans exhibited only minor differences, and marmosets showed no differences in eye movements between the Random and GD animations, coupled with task design constraints, we only used the Random and ToM animations for the fMRI studies in both humans and marmosets (see Materials and methods).

We first investigated ToM and Random animations processing in humans. *Figure 3* shows group activation maps for ToM (A) and Random (B) conditions as well as the comparison between ToM and Random conditions (C) obtained for human participants.

Both ToM (*Figure 3A*) and Random (*Figure 3B*) videos activated a large bilateral network. While the same larger areas were activated in both conditions, the specific voxels showing this activation within those areas were typically distinct. In some cases, both conditions activated the same voxels, but the degree of activation differed. This suggests a degree of both spatial and intensity variation in the activations for the two conditions within the same areas. The activated areas included visual areas (V1, V2, V3, V3CD, V3B, V4, V4T, V6A, V7, MT, MST), lateral occipital areas 1, 2, and 3 (LO1, LO2, LO3), temporal areas (FST, PH, PHT, TE2, posterior inferotemporal complex PIT and fusiform face complex FFC), temporo-parietal junction areas (TPOJ2 and TPOJ3), lateral posterior parietal areas also comprising the parietal operculum (supramarginal areas PF, PFt, angular areas PGp and PGi, superior temporal visual area STV, perisylvian language area PSL, medial intraparietal area MIP, ventral and dorsal lateral intraparietal areas LIPv and LIPd, anterior intraparietal area AIP, IPS1, IPS0, 7PC and 5 L), medial superior parietal areas (7am, PCV, 5 mv), secondary somatosensory cortex (S2), premotor areas (6, 55b, premotor eye field PEF, frontal eye field FEF), and frontal areas (8Av, 8 C, IFJp, IFla).

The ToM condition (*Figure 3A*) also showed bilateral activations in posterior superior temporal sulcus (STSdp), in temporo-parietal junction area TPOJ1, in ventral visual complex (VVC), in parahippocampal area 3 (PHA3), in lateral posterior parietal areas Pfop and PFcm, in lateral prefrontal areas 8 C, 8Av, 44 and 45, in inferior frontal areas IFSp, IFSa, and in frontal opercular area 5 (FOP5).

To identify brain areas that are more active during the observation of ToM compared to Random videos, we directly compared the two conditions (i.e., ToM animations >Random animations contrast, *Figure 3C* and Figure 5A). This analysis reveals increased activations for the ToM condition compared to the Random condition in occipital, temporal, parietal and frontal areas. This includes notable differences in the bilateral visual areas V1, V2, V3, V3CD, V4, V4t, MT, MST, as well as the bilateral LO1, LO2 and LO3 regions. The increase extends into the lateral temporal lobe, as observed in the bilateral

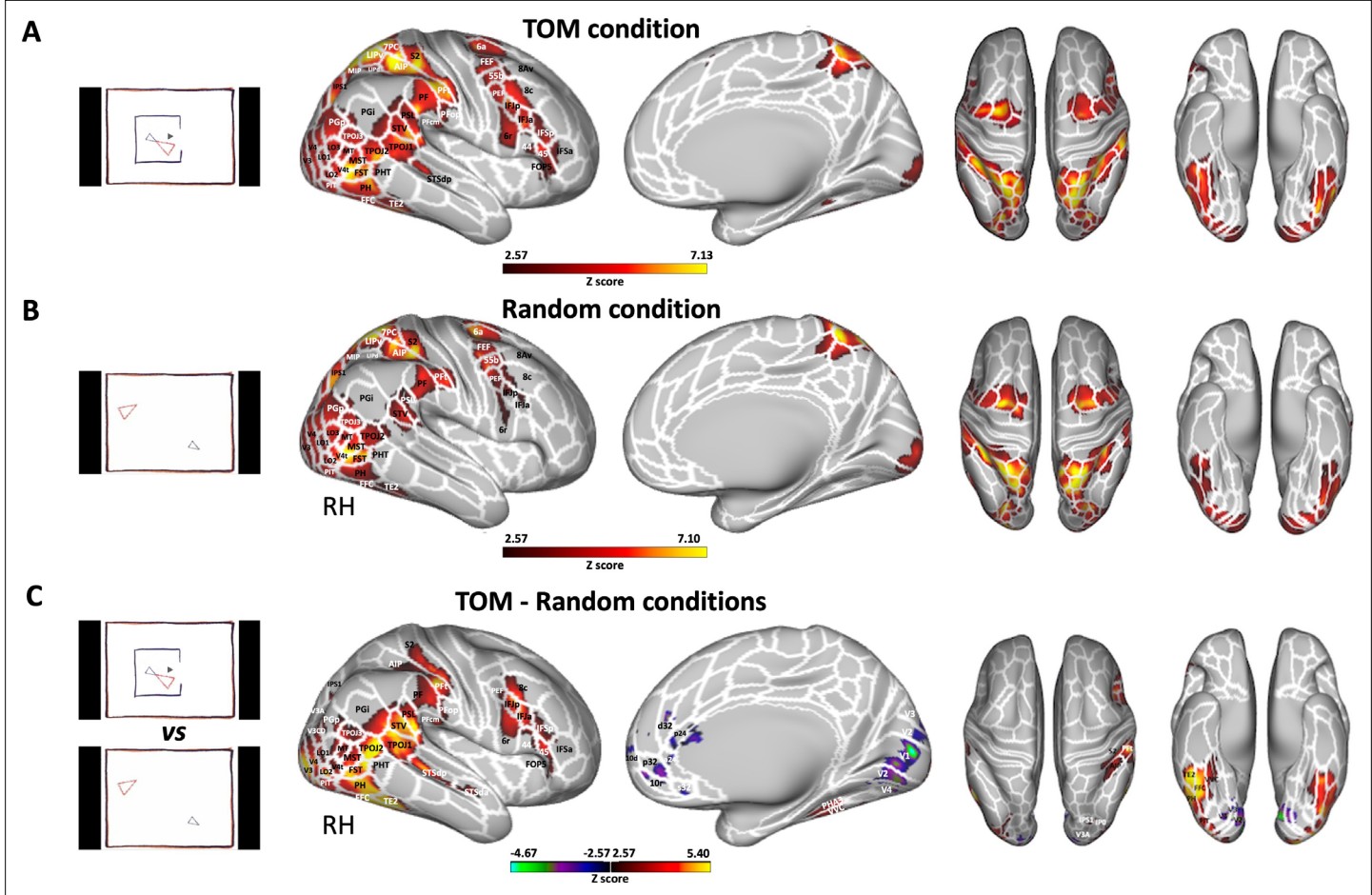

**Figure 3.** Brain networks involved in processing of Frith-Happé's ToM and Random animations in humans. Group functional maps displayed on right fiducial (lateral and medial views) and left and right fiducial (dorsal and ventral views) of human cortical surfaces showing significant greater activations for ToM condition (**A**), Random condition (**B**) and the comparison between ToM and Random conditions (**C**). The white line delineates the regions based on the recent multi-modal cortical parcellation atlas (*Glasser et al., 2016*). The maps depicted are obtained from 10 human subjects with an activation threshold corresponding to z-scores >2.57 for regions with yellow/red scale or z-scores <–2.57 for regions with purple/green scale (AFNI's 3dttest++, cluster-forming threshold of p<0.01 uncorrected and then FWE-corrected α=0.05 at cluster-level from 10000 Monte-Carlo simulations).

The online version of this article includes the following figure supplement(s) for figure 3:

**Figure supplement 1.** Brain networks involved in ToM animations processing in humans.

**Figure supplement 2.** Image and temporal signal-to-noise-ration (SNR) calculated on fMRI data acquired at 7T with an AC-84 Mark II gradient coil, an in-house 8-channel transmit, and a 32-channel receive coil (see methods, main text).

PH and FST areas and into the more inferior part of the temporal lobe in the bilateral TE2, FFC, and PIT areas. We also found greater activations in the bilateral temporo-parietal junction areas (TPOJ1, TPOJ2, TPOJ3), and along the right STS in STSdp and STSda areas. This extended to left and right parietal areas, especially in the inferior parietal lobule, including the right supramarginal and opercular supramarginal areas (PF, PFm, PFt, Pfop, and PFcm), left PFt, bilateral opercular areas PSL and STV, bilateral angular areas PGp and Pgi, right IPS1, and bilateral IP0. The activation also extended into the superior parietal lobule (right AIP). Moving anteriorly, we observed greater activations during ToM animations in the secondary somatosensory cortex, premotor areas (6 r and PEF), lateral prefrontal areas (8 C, 44, and 45), and the inferior (IFSa, IFSp, IFJa, and IFJp) as well as the opercular (FOP5) frontal areas in the right hemisphere. In contrast, the Random condition exhibited greater activations, than the ToM condition predominantly within the left and right visual areas (V1, V2, V3, V3A, V4) and in dorsolateral (10d and 10r bilateral), lateral (9m left) and medial frontal areas (d32 and a24 bilateral, p24 and s32 right).

At the subcortical level (see *Figure 5—figure supplement 1*, left panel), we observed enhanced bilateral activations in the cerebellum and in certain areas of the thalamus (namely, the right ventroposterior thalamus or THA-VP, and the left and right dorsoanterior thalamus or THA-DA) under both ToM (*Figure 5—figure supplement 1A*, left panel) and Random conditions (*Figure 5—figure supplement 1B*, left panel) when compared to the baseline. Additionally, a small section of the right amygdala was engaged in the ToM condition. We noted more pronounced activations in the posterior lobe of the cerebellum, the right amygdala and thalamus (right THA-VP, right ventroanterior thalamus or THA-VA, and left and right dorsoposterior thalamus or THA-DP) for the ToM condition compared to the Random condition (*Figure 5—figure supplement 1C*, left panel). No regions showed greater activations for Random condition compared to the ToM condition.

As we used shorter modified versions of the Frith-Happé animations (i.e. videos of 19.5 s instead of 40 s), we also validated our stimuli and our fMRI protocol by comparing the brain responses elicited by ToM animation videos – compared to Random animation videos – obtained in our group of 10 human subjects and those reported by the large group of humans (496) used in the social cognition task of the Human Connectome Project (HCP; *Barch et al., 2013*), which also used shortened versions of the Frith-Happé animations.

This comparison is shown in *Figure 3—figure supplement 1*. Overall, we observed similar distinct patterns of brain activations (*Figure 3—figure supplement 1A and B*), including a set of areas in occipital, temporal, parietal and frontal cortices, as described previously (*Figure 3C*). The main differences were stronger activations in the left hemisphere in the HCP dataset. Therefore, these results

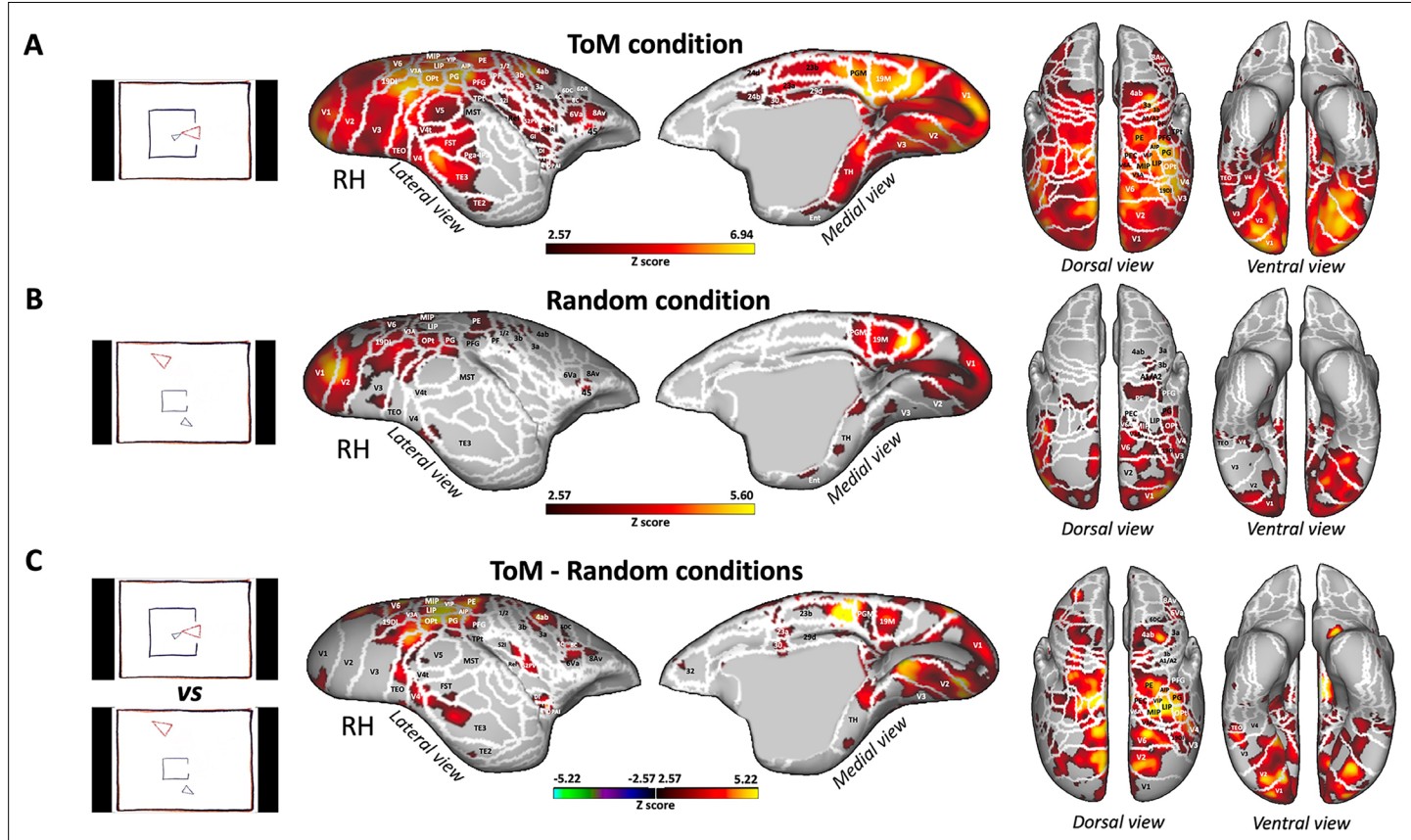

**Figure 4.** Brain networks involved in processing of Frith-Happé's ToM and Random animations in marmosets. Group functional maps showing significant greater activations for ToM condition (**A**), Random condition (**B**) and the comparison between ToM and Random conditions (**C**). Group map obtained from six marmosets displayed on lateral and medial views of the right fiducial marmoset cortical surfaces as well as dorsal and ventral views of left and right fiducial marmoset cortical surfaces. The white line delineates the regions based on the Paxinos parcellation of the NIH marmoset brain atlas (*Liu et al., 2018*). The brain areas reported have activation threshold corresponding to z-scores >2.57 (yellow/red scale) or z-scores <–2.57 (purple/green scale) (AFNI's 3dttest++, cluster-forming threshold of p<0.01 uncorrected and then FWE-corrected α=0.05 at cluster-level from 10,000 Monte-Carlo simulations).

show that our stimuli and our protocol are appropriate to investigate mental state attribution to animated moving shapes.

## Functional brain activations while watching ToM and Random Frith-Happé's animations in marmosets

Having identified the brain regions activated during the processing of ToM or Random videos in human subjects and validated our protocol, we proceeded to use the same stimuli in marmosets. *Figure 4* illustrates the brain network obtained for the ToM condition (A), Random condition (B), and the contrast between ToM and Random conditions (C) in six marmosets.

Both the ToM (*Figure 4A*) and Random (*Figure 4B*) animations activated an extensive network involving a variety of areas in the occipito-temporal, parietal and frontal regions. As in human subjects, it should be noted that while both conditions elicited strong activation in some of the same larger areas, these activations might have either occurred in distinct voxels within those areas, or the same voxels were activated to varying degrees for both conditions. This suggests distinct yet overlapping patterns of neural processing for the ToM and Random conditions.

In the occipital and temporal cortex, the activations were located in the visual areas V1, V2, V3, V3A, V4, V4t, V5, V6, MST, the medial (19 M), and dorsointermediate parts (19DI) of area 19, ventral temporal area TH, enthorinal cortex, and lateral and inferior temporal areas TE3 and TEO. Activations were also observed in the posterior parietal cortex, specifically in bilateral regions surrounding the intraparietal sulcus (IPS), in areas LIP, MIP, PE, PG, PFG, PF, V6A, PEC, in the occipito-parietal transitional area (OPt) and in medial part of the parietal cortex (area PGM). More anteriorly, bilateral activations were present in areas 1/2, 3a, 3b of the somatosensory cortex, in primary motor area 4 parts a, b and c (area 4ab and 4c), in area 6 ventral part (6Va) of the premotor cortex and in frontal areas 45 and 8Av.

The ToM condition (*Figure 4A*) also recruited bilateral activations in areas V5, TE2, FST, Pga-IPa, temporoparietal transitional area (TPt), around the IPS in AIP and VIP, in the internal part (S2I), parietal rostral part (S2PR) and ventral part (S2PV) of the secondary somatosensory cortex, in agranular insular cortex (AI), granular and dysgranular insular areas (GI and DI), retroinsular area (ReI) and orbital peri-allocortex (OPAI), as well as in premotor cortex in area 8 caudal part (8C), in area 6 dorsocaudal and dorsorostral parts (6DC, 6DR). Additionally, we also observed activations in posterior cingulate areas 23a, 23b, 29d, 30, 24d, and 24b.

Next, we examined the difference between ToM and Random animations (i.e. ToM condition >Random condition contrast, *Figure 4C* and *Figure 5B*). We found enhanced bilateral activations for the ToM condition across a range of regions. These encompassed occipital areas V1, V2, V3, V3A, V4, V4t, V5, V6, 19DI, 19M, temporal areas TH, TE2, TE3, FST, MST, TPt, and parietal areas LIP, MIP, VIP, AIP, PE, PG, PFG, OPt, V6A, PEC. Moreover, these activations extended to the somatosensory cortex (areas 1/2, 3 a, 3b, S2I, S2PV), the primary motor cortex (areas 4ab and 4c), lateral frontal areas 6DC, 8C, 6Va, 8Av, 8Ad (left hemisphere), and insular areas (ReI, S2I, S2PV, DI, AI). Additional activations were observed in the OPAI area, medial frontal area 32 and posterior cingulate areas (23 a, 23b, 29d, 30). Contrarily, we did not find any regions exhibiting stronger activations for the Random condition compared to the ToM condition. This further emphasizes the distinctive neural recruitment and processing associated with ToM animations within the marmoset brain.

At the subcortical level (see *Figure 5—figure supplement 1A*, right panel), the ToM condition showed involvement of several areas including the bilateral hippocampus, bilateral pulvinar (lateral, medial and inferior parts), bilateral amygdala, and left caudate. On the other hand, the Random condition recruited only the pulvinar (*Figure 5—figure supplement 1B*, right panel). Upon comparison of the ToM and Random conditions, the ToM animations showed stronger activations in the right superior colliculus (SC), right lateral geniculate nucleus (LGN), left caudate, left amygdala, left hippocampus and certain portions of the right and left pulvinar (lateral and inferior pulvinar; *Figure 5—figure supplement 1C*, right panel).

## Comparison of functional brain activations in humans and marmosets

As described earlier, both humans (*Figure 5A*) and marmosets (*Figure 5B*) displayed an extended network of activations across the occipital, temporal, parietal, and frontal cortices in response to ToM animations compared to Random animations. Overall, there were substantial similarities between the

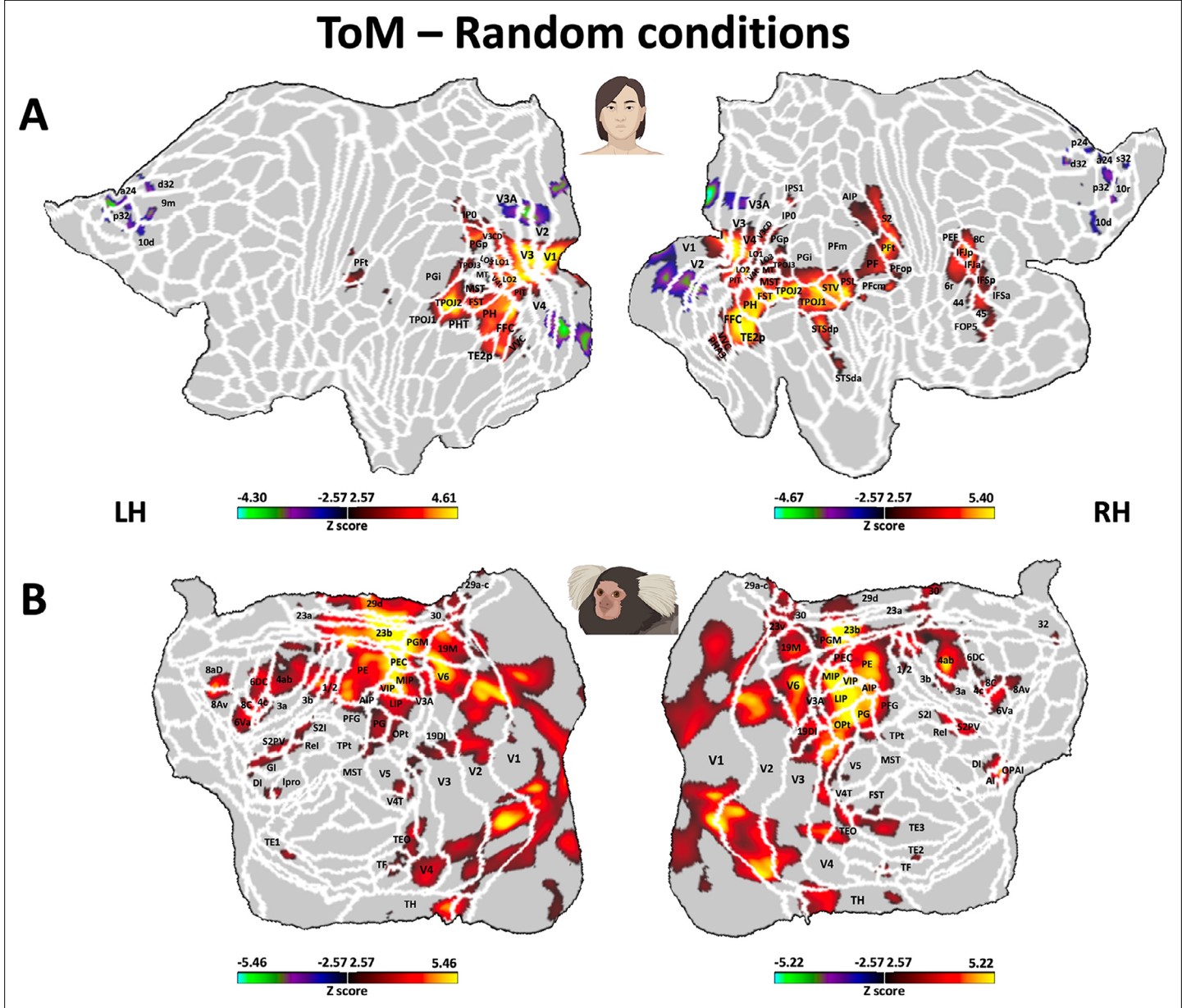

**Figure 5.** Brain network involved during processing of ToM compared to Random Frith-Happé's animations in both humans (**A**) and marmosets (**B**). Group functional maps showing significant greater activations for ToM animations compared to Random animations. (**A**) Group map obtained from 10 human subjects displayed on the left and right human cortical flat maps. The white line delineates the regions based on the recent multi-modal cortical parcellation atlas (*Glasser et al., 2016*). (**B**) Group map obtained from 6 marmosets displayed on the left and right marmoset cortical flat maps. The white line delineates the regions based on the Paxinos parcellation of the NIH marmoset brain atlas (*Liu et al., 2018*). The brain areas reported in A and B have activation threshold corresponding to z-scores >2.57 (yellow/red scale) or z-scores <−2.57 (purple/green scale) (AFNI's 3dttest++, cluster-forming threshold of p<0.01 uncorrected and then FWE-corrected α=0.05 at cluster-level from 10,000 Monte-Carlo simulations).

The online version of this article includes the following figure supplement(s) for figure 5:

**Figure supplement 1.** Subcortical activations during processing of Frith-Happé's ToM and Random animations in humans (left) and marmosets (right).

two species, with both exhibiting enhanced activations for ToM animations compared to Random animations within visual areas, inferior and superior temporal areas, the inferior parietal lobe, and AIP area encircling the IPS in the superior parietal lobe. We also found parallel activations in the somatosensory cortex, although the activation was more widespread in marmosets compared to humans, where it was confined to the secondary somatosensory cortex. Additional similarities were identified in the premotor cortex and certain regions of the lateral prefrontal cortex. Overall, left and

right hemisphere activations demonstrated greater congruity in marmosets compared to humans. However, this might be attributed to our human head coil, which had a lower signal-to-noise ratio (SNR) in the left hemisphere (see *Figure 3—figure supplement 2*). Indeed, similar bilateral activations in humans have been observed in the human HCP dataset (*Barch et al., 2013*; see *Figure 3—figure supplement 1B*).

Nevertheless, there were also discernible differences between the two species for ToM compared to Random animations, including stronger activations in medial frontal cortex, primary motor area and posterior cingulate cortex for marmosets, which were absent in our human sample. Moreover, different parts of the insular cortex were recruited in marmosets, whereas in humans, activations were limited to the parietal operculum and did not extend into the insula. At the subcortical level, although both humans and marmosets demonstrated activations in the amygdala, humans recruited the dorsal thalamus and the cerebellum, whereas marmosets displayed activations in the hippocampus, the SC and the LGN.

These results indicate that, while there were many shared brain activation patterns in both humans and marmosets during the processing of ToM animations compared to Random animations, several notable species-specific differences were also evident.

## Discussion

In the present study, we investigated whether New-World common marmoset monkeys, like humans, process videos of animated abstract shapes differently when these shapes appear to be reacting to each other (ToM condition) compared to when they interact in a purely physical manner (GD condition) or when they move purposelessly (Random condition). To facilitate a direct comparative analysis between the two primate species, we measured their gaze patterns and brain activations as they viewed the widely-used Frith-Happé's animations (*Abell et al., 2000*; *Castelli et al., 2000*). In these animations, the ToM condition is characterized by one triangle reacting to the other's mental state, exemplifying behaviors like coaxing, surprising, seducing, and mocking. In the GD condition, the two triangles appear to engage purely physically without any implied mental attribution, depicting behaviors such as chasing, dancing, fighting, and leading. In the Random condition, the two triangles move independently, depicting motions akin to a game of billiards, drifting movements, a star pattern, or a tennis game. In all these animations, the physical interaction of the triangles does appear to follow the laws of physics in a reasonably predictable manner. This is probably most evident in the random 'billiard' condition in which the two triangles bounce off the walls. However, the ToM animations also follow Newton's third law, for example when the small triangle is trying to get inside the box and bounces against it in the 'seducing' condition, or when the large triangle pushes the small triangle in the 'coaxing' condition.

In our first experiment, we examined the gaze patterns of marmosets and humans during the viewing of these video animations. *Klein et al., 2009* reported differing fixation durations for these animations, where the longest fixations were observed for ToM animations, followed by GD animations and the shortest fixations for Random animations. They further reported that the intentionality score - derived from verbal descriptions of the animations - followed a similar pattern: highest for ToM, lowest for Random, and intermediate for GD animations. This validated the degree of mental state attribution according to the categories and established that animations provoking mentalizing (ToM condition) were associated with long fixations. This, in turn, supports the use of fixation durations as a nonverbal metric for mentalizing capacity (*Klein et al., 2009*; *Meijering et al., 2012*). Our results with human subjects, which demonstrated longer fixation durations for the ToM animations compared to the GD and Random animations, paralleled those of *Klein et al., 2009*. However, unlike Klein et al.'s findings, we did not observe intermediate durations for GD animations in our study.

Interestingly, our marmoset data did not align with the human findings but instead resonated more with *Schafroth et al., 2021*'s observations in macaque monkeys, which did not show significant differences in fixation durations across the three animation types.

Our study went a step further than previous research in humans (*Klein et al., 2009*) and macaques (*Schafroth et al., 2021*) by investigating the proportion of time that subjects devoted to looking at the two central figures in the animations: the large red triangle and the small blue triangle. Our results

indicate that both humans and marmosets spent significantly more time looking at the large red triangle during ToM, compared to GD and Random animations. Humans also exhibited a preference for the red triangle in the GD over the Random condition, a differentiation not evident in marmosets. This result suggests that marmosets process these conditions similarly, indicating that, unlike humans, they do not seem to discern a marked difference between purposeless and goal-directed motions. However, they did show a distinctive gaze pattern in the ToM condition, pointing to their capacity to potentially perceive or react to animated sequences with complex mental interactions. Our findings revealed no significant differences in gaze patterns towards the smaller blue triangle across the three conditions in both humans and marmosets, potentially due to the perception of the large red triangle as a more salient or socially relevant figure in the interactions.

Together, the observed gaze patterns do not support the idea that marmosets increase their cognitive processing during ToM animations in the same way as humans. However, the findings point to a certain level of sophistication in the marmosets' perception of abstract ToM animations.

Thus, in our second experiment, we investigated the brain networks involved when viewing the ToM and Random Frith-Happé's animations in humans and marmosets. Previous fMRI studies in humans identified a specific network associated with ToM processing in tasks such as stories, humorous cartoons, false-belief tasks, and social gambling. This network typically includes areas such as the medial frontal gyrus, posterior cingulate cortex, inferior parietal cortex, and temporoparietal junction (*Fletcher et al., 1995*; *Gallagher et al., 2000*). Nevertheless, these studies used complex stimuli and yielded heterogeneous results, with varied activations in regions such as the medial and lateral prefrontal cortex, inferior parietal lobule, occipital cortex, and insula across different studies (*Carrington and Bailey, 2009*). This variability is likely due to the diverse experimental paradigms employed to study ToM. Studies employing Frith-Happé's animations, which are less complex and more controlled, reported distinct patterns of brain activation when viewing ToM compared to Random animations, involving areas such as the dorso-medial prefrontal cortex, inferior parietal cortex, temporoparietal, inferior and superior temporal regions, and lateral superior occipital regions (*Barch et al., 2013*; *Bliksted et al., 2019*; *Castelli et al., 2000*; *Chen et al., 2023*; *Gobbini et al., 2007*; *Vandewouw et al., 2021*; *Weiss et al., 2021*; *Wheatley et al., 2007*).

Our slightly adapted versions of the Frith-Happé animations led to a similar distinct pattern of brain activations, with an exception for the lack of activations in the dorsal part of the medial prefrontal cortex. This discrepancy could be attributable to various factors, including differences in task design, methodological aspects such as statistical power, or variations in participants characteristics. Otherwise, our results and those of the HCP data from *Barch et al., 2013*, revealed stronger activations for ToM versus Random animations in areas in premotor, prefrontal, parietal, visual, inferior and superior temporal cortices including the STS and temporoparietal junction. Overall, the ToM network we identified, as well as that reported by *Barch et al., 2013*, appear to be more extensive than those described in studies employing more complex experimental paradigms to study ToM. This aligns with the recent meta-analysis conducted by *Schurz et al., 2021*, which demonstrated that the network activated by simpler, non-verbal stimuli like social animations differs from the traditional network, with involvement of both cognitive and affective networks (*Schurz et al., 2021*).

As in humans, the comparison of responses to ToM and Random animations in marmosets revealed activations in occipito-temporal, parietal, and frontal regions. Specifically, activations in the TE areas in marmosets could be equivalent to those observed along the STS in humans (*Yovel and Freiwald, 2013*). We also observed in both our human subjects and marmosets, activations in the inferior parietal cortex, previously reported in human literature (*Carrington and Bailey, 2009*; *Fletcher et al., 1995*; *Gallagher et al., 2000*). We also found the similar activations in the superior parietal cortex in both our human and marmoset subjects, specifically in the area surrounding the IPS, but this have not been predominantly described in previous work (*Carrington and Bailey, 2009*; *Chen et al., 2023*; *Fletcher et al., 1995*; *Gallagher et al., 2000*; *Gobbini et al., 2007*). However, there are also noteworthy differences between our results and those of our human data. Firstly, while we did not observe activations in the medial prefrontal cortex in humans, they were present in marmosets, aligning with previous human fMRI studies (*Bliksted et al., 2019*; *Castelli et al., 2000*; *Weiss et al., 2021*; *Wheatley et al., 2007*). The marmoset network also included the posterior cingulate cortex and the

insula, areas known to be involved in mentalizing and affective processing respectively in human ToM studies that employed more complex stimuli (*Fletcher et al., 1995*; *Gallagher et al., 2000*; *Wheatley et al., 2007*). Finally, a prominent difference between humans and marmosets is the strong activation in the marmoset motor cortex for ToM animations, which was absent in humans, in addition to the differences observed at the subcortical level. Interestingly, we have also recently reported activations in marmoset primary motor cortex during the observation of social interactions (*Cléry et al., 2021*), suggesting a potential role for the marmoset motor cortex in interaction observation. Regarding the distinct subcortical activations observed in humans and marmosets, it's important to consider the specific social cognitive demands that might be unique to each species. The involvement of the dorsal thalamus, cerebellum, and a small portion of the amygdala in humans may reflect the complexities of information processing, social cognition, and emotional involvement required to interpret the ToM animations (e.g. *Halassa and Sherman, 2019*; *Janak and Tye, 2015*; *Van Overwalle et al., 2014*). Conversely, the activation of the amygdala and hippocampus in marmosets could suggest a more emotion- and memory-based processing of the social stimuli (e.g. *Eichenbaum, 2017*; *Van Overwalle et al., 2014*). However, it's critical to consider that these interpretations are speculative and would require further study for confirmation.

Together, these findings demonstrate that marmosets, while observing interacting animated shapes as opposed to randomly moving shapes, exhibit enhanced activation in several brain regions previously associated with ToM processing in humans.

Interestingly, our results differed from those obtained by *Roumazeilles et al., 2021* in their fMRI study conducted in macaques using the same animations. Roumazeilles and colleagues reported no differences in activation between ToM and Random animations, suggesting that rhesus macaques may not respond to the social cues presented by the ToM Frith-Happé animations. This disparity between our marmoset findings and those of macaques raises intriguing questions about potential differences in the evolutionary development of ToM processing within non-human primates. Marmosets, as New World monkeys, are part of an evolutionary lineage that diverged earlier than the lineage of Old-World monkeys such as macaques. This difference in lineage might lead to distinct evolutionary trajectories in cognitive processing, which could include varying sensitivity to abstract social cues in animations.

In summary, our study reveals novel insights into how New World marmosets, akin to humans, differentially process abstract animations that depict complex social interactions and animations that display purely physical or random movements. Our findings, supported by both specific gaze behaviors (i.e. the proportion of time spent on the red triangle, despite the inconclusiveness of overall fixation) and distinct neural activation patterns, shed light on the marmosets' capacity to interpret social cues embedded in these animations.

The differences observed between humans, marmosets, and macaques underscore the diverse cognitive strategies that primate species have evolved to decipher social information. This diversity may be influenced by unique evolutionary pressures that arise from varying social structures and lifestyles. Like macaque monkeys, humans often live in large, hierarchically organized social groups where status influences access to resources. However, both humans and marmosets share a common trait: a high degree of cooperative care for offspring within the group, with individuals other than the biological parents participating in child-rearing. These distinctive social dynamics of marmosets and humans may have driven the development of unique social cognitive abilities. This could explain their enhanced sensitivity to abstract social cues in the Frith-Happé animations.

Nonetheless, it is crucial to emphasize that even though marmosets respond to the social cues in the Frith-Happé animations, this does not automatically imply that they possess mental-state attributions comparable to humans. As such, future research including a range of tasks, from sensory-affective components to more abstract and decoupled representations of others' mental states (*Schurz et al., 2021*), will be fundamental in further unravelling the complexities of the evolution and functioning of the Theory of Mind across the primate lineage.

## Materials and methods

### Common marmosets

All experimental procedures were in accordance with the Canadian Council of Animal Care policy and a protocol approved by the Animal Care Committee of the University of Western Ontario Council on Animal Care #2021–111.

Eleven adult marmosets (4 females, 32–57 months, mean age: 36.6 months) were subjects in this study. All animals were implanted for head-fixed experiments with either a fixation chamber (*Johnston et al., 2018*) or a head post (*Gilbert et al., 2023*) under anesthesia and aseptic conditions. Briefly, the animals were placed in a stereotactic frame (Narishige, model SR-6C-HT) while being maintained under gas anaesthesia with a mixture of $O_2$ and air (isoflurane 0.5–3%). After a midline incision of the skin along the skull, the skull surface was prepared by applying two coats of an adhesive resin (All-Bond Universal; Bisco, Schaumburg, IL) using a microbrush, air-dried, and cured with an ultraviolet dental curing light (King Dental). Then, the head post or fixation chamber was positioned on the skull and maintained in place using a resin composite (Core-Flo DC Lite; Bisco). Heart rate, oxygen saturation, and body temperature were continuously monitored during this procedure.

Six of these animals (four females - weight 315–442 g, age 30–34 months - and two males - weight 374–425 g, age 30 and 55 months) were implanted with an MRI-compatible machined PEEK head post (*Gilbert et al., 2023*). Two weeks after the surgery, these marmosets were acclimatized to the head-fixation system in a mock MRI environment.

### Human participants

Eleven healthy humans (4 females, 25–42 years, mean age: 30.7 years) participated in the eye tracking experiment. Among these, five individuals, along with five additional subjects (4 females, 26–45 years), took part in the fMRI experiment. All subjects self-reported as right-handed, had normal or corrected-to-normal vision and had no history of neurological or psychiatric disorders. Importantly, all subjects confirmed they had not previously been exposed to the Frith-Happé animation videos used in our study. Subjects were informed about the experimental procedures and provided informed written consent. These studies were approved by the Ethics Committee of the University of Western Ontario.

### Stimuli

Eight animations featuring simple geometric shapes with distinct movement patterns were used (*Figure 1*). These animations, originally developed by Abell and colleagues (*Abell et al., 2000*), presented two animated triangles - a large red triangle and a small blue triangle - moving within a framed white background. The original social animation task included three conditions: ToM, Goal-Directed (GD), and Random. In the ToM animations, one triangle displayed behaviors indicative of mental interactions by reacting to the mental state of the other triangle. The GD animations depicted simple interactions between the two triangles, while the Random animations showed the triangles moving and bouncing independently.

The ToM animations portrayed various scenarios, such as one triangle attempting to seduce (*Video 1*) or persuade the other mocking it behind its back (*Video 2*), surprising it by hiding behind a door (*Video 3*), or coaxing it out of an enclosure (*Video 4*). In the GD animations, the triangles could

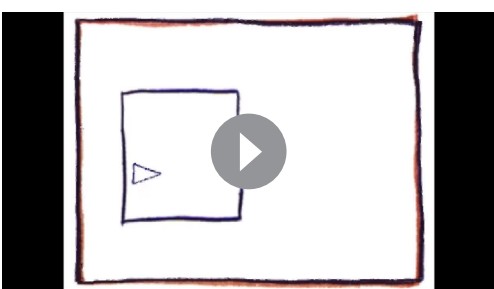

**Video 1.** Theory of Mind (ToM)' Category, Frith-Happe Animations – Seducing Simulation.
https://elifesciences.org/articles/86327/figures#video1

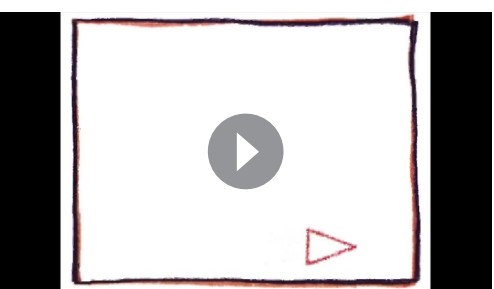

**Video 2.** Theory of Mind (ToM)' Category, Frith-Happe Animations – Mocking Simulation.
https://elifesciences.org/articles/86327/figures#video2

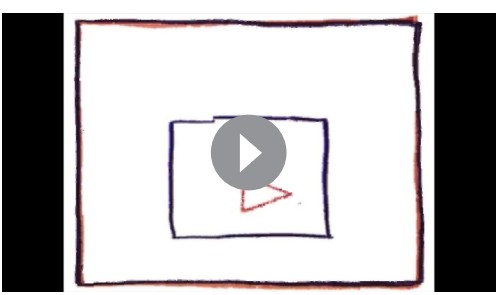

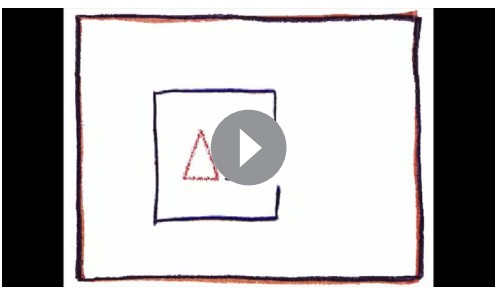

**Video 3.** Theory of Mind (ToM)' Category, Frith-Happe Animations – Surprise Simulation.
https://elifesciences.org/articles/86327/figures#video3

**Video 4.** Theory of Mind (ToM)' Category, Frith-Happe Animations – Coaxing Simulation.
https://elifesciences.org/articles/86327/figures#video4

dance together (*Video 5*), fight together (*Video 6*), or one triangle could chase (*Video 7*) or lead the other (*Video 8*). The Random animations featured independent movements of the triangles, following patterns such as billiard (*Video 9*), drifting (*Video 10*), star (*Video 11*), or tennis (*Video 12*). Similar to the approach used in the HCP study (*Barch et al., 2013*), we modified the original video clips and shortened each animation to 19.5 s using custom video-editing software (iMovie, Apple Incorporated, CA).

## Eye tracking task and data acquisition

To investigate potential behavioral differences during the viewing of Frith-Happé animations, we presented all ToM, GD and Random video clips once each in a pseudorandomized manner to both marmoset and human subjects. The presentation of stimuli was controlled using Monkeylogic software (*Hwang et al., 2019*). All stimuli were presented on a CRT monitor (ViewSonic Optiquest Q115, 76 Hz non-interlaced, 1600 x 1280 resolution). Eye position was digitally recorded at 1 kHz via video tracking of the left pupil (EyeLink 1000, SR Research, Ottawa, ON, Canada).

At the beginning of each session, horizontal and vertical eye positions of the left eye were calibrated by presenting a 1 degree dot at the display centre and at 6 degrees in each of the cardinal directions for 300–600ms. Monkeys were rewarded at the beginning and end of each session. Crucially, no rewards were provided during the calibration or while the videos were played.

## fMRI task

For the fMRI experiment, it was crucial for us to ensure that the subjects remained alert and focused throughout the entire scanning session, which becomes increasingly difficult with longer runs. There, we used only the ToM and Random conditions in our functional runs, as the GD condition is situated between these two extremes, depicting physical interaction among the triangles without suggesting any mental state attribution. The limitation to ToM and random conditions is consistent with the design of previous fMRI studies in humans and macaques that employed Frith-Happé animations

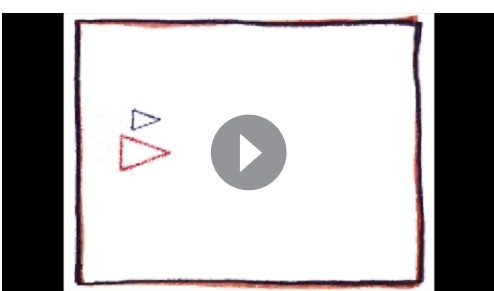

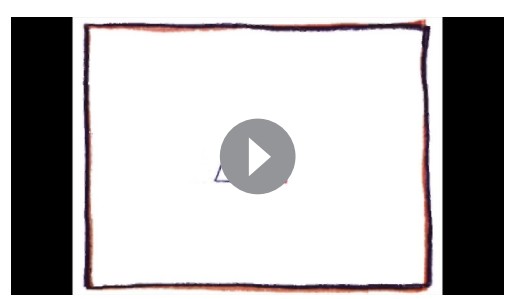

**Video 5.** Goal-Directed (GD)' Category, Frith-Happe Animations – Dancing Simulation.
https://elifesciences.org/articles/86327/figures#video5

**Video 6.** Goal-Directed (GD)' Category, Frith-Happe Animations – Fighting Simulation.
https://elifesciences.org/articles/86327/figures#video6

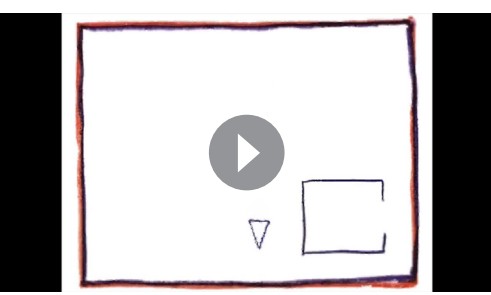

**Video 7.** Goal-Directed (GD)' Category, Frith-Happe Animations – Chase Simulation.
https://elifesciences.org/articles/86327/figures#video7

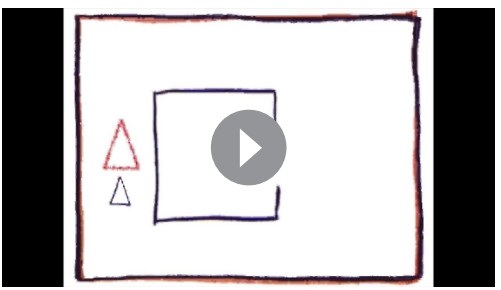

**Video 8.** Goal-Directed (GD)' Category, Frith-Happe Animations – Leading Simulation.
https://elifesciences.org/articles/86327/figures#video8

(*Gobbini et al., 2007*; *Barch et al., 2013*; *Bliksted et al., 2019*; *Vandewouw et al., 2021*; *Weiss et al., 2021*; *Chen et al., 2023*; *Roumazeilles et al., 2021*).

Humans and marmosets were presented with ToM and Random video clips in a block design. Each run consisted of eight blocks of stimuli (19.5 s each) interleaved by baseline blocks (15 s each). ToM or Random animations were presented pseudorandomly, and each condition was repeated four times (*Figure 1*). For each run, the order of these conditions was randomized leading to 14 different stimulus sets, counterbalanced within and between subjects. In baseline blocks, a 0.36° circular black cue was displayed at the center of the screen against a gray background. We found previously that such a stimulus reduced the vestibulo-ocular reflex evoked by the strong magnetic field.

## fMRI experimental setup

During the scanning sessions, the marmosets sat in a sphinx position in a custom-designed plastic chair positioned within a horizontal magnet (see below). Their head was restrained using a head fixation system allowing to secure the surgically implanted head post to a clamping bar (*Gilbert et al., 2023*). After the head was immobilized, the two halves of the coil housing were positioned on either side of the head. Inside the scanner, monkeys faced a translucent screen placed 119 cm from their eyes where visual stimuli were projected with an LCSD-projector (Model VLP-FE40, Sony Corporation, Tokyo, Japan) via a back-reflection on a first surface mirror. Visual stimuli were presented with the Keynote software (version 12.0, Apple Incorporated, CA) and were synchronized with MRI TTL pulses triggered by a Raspberry Pi (model 3B+, Raspberry Pi Foundation, Cambridge, UK) running via a custom-written Python program. No reward was provided to the monkeys during the scanning sessions. Animals were monitored using an MRI-compatible camera (Model 12M-I, MRC Systems GmbH). Horizontal and vertical eye movements were monitored at 60 Hz using a video eye tracker (ISCAN, Boston, Massachusetts). While we were able to obtain relatively stable eye movement recordings from a few runs per animal (min 1, max 5 runs per animal), the quality of the recordings was not sufficient for a thorough analysis. The large marmoset pupil represents a challenge for video eye tracking when the eyes are not fully open. Data from functional runs with more stable eye signals (n=15) show good compliance

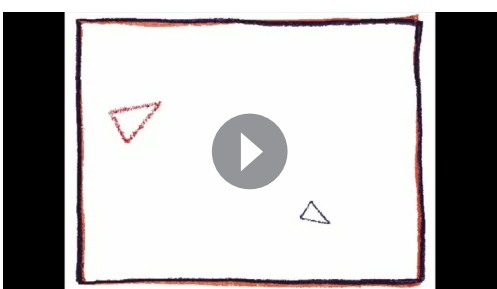

**Video 9.** Random' Category, Frith-Happé Animations – Billiard Simulation.
https://elifesciences.org/articles/86327/figures#video9

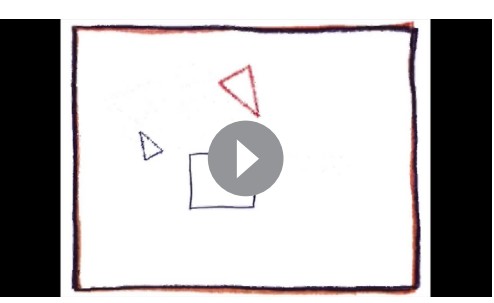

**Video 10.** Random' Category, Frith-Happé Animations – Drifting Simulation.
https://elifesciences.org/articles/86327/figures#video10

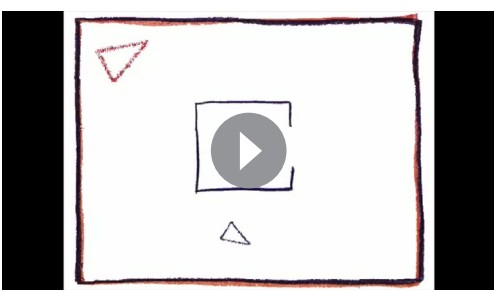

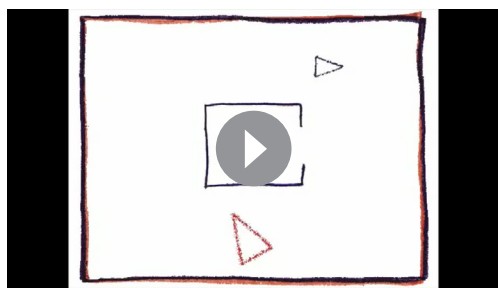

**Video 11.** Random' Category, Frith-Happé Animations – Star Simulation.
https://elifesciences.org/articles/86327/figures#video11

**Video 12.** Random' Category, Frith-Happé Animations – Tennis Simulation.
https://elifesciences.org/articles/86327/figures#video12

in the marmosets. The percentage of time spent in each run looking at the screen in the two experimental conditions (ToM, Random) and during the Baseline periods (fixation point in the center of the screen) was higher than 85% (88.2%, 88.6% and 93.4% respectively for ToM, Random and Baseline conditions). There was no significant differences between the ToM and Random condition (paired t-test, $t_{(14)}$=-0.374, p=0.71), ruling out the possibility that any differences in fMRI activation between the ToM and Random condition were simply due to a different exposure to the videos.

Human subjects lay in a supine position and watched the stimuli presented via a rear projection system (Avotech SV-6011, Avotec Incorporated) through a surface mirror affixed to head coil. As for marmosets, visual stimuli were presented with the Keynote software (version 12.0, Apple Incorporated, CA) and were synchronized with MRI TTL pulses triggered by a Raspberry Pi (model 3B+, Raspberry Pi Foundation, Cambridge, UK) running via a custom-written python program.

## MRI data acquisition

Marmoset and human imaging were performed at the Center for Functional and Metabolic Mapping at the University of Western Ontario.

For marmoset subjects, fMRI data were acquired on a 9.4T 31 cm horizontal bore magnet (Varian) with a Bruker BioSpec Avance III HD console running software package Paravision-360 (Bruker BioSpin Corp), a custom-built high-performance 15 cm diameter gradient coil (maximum gradient strength: 1.5 mT/m/A), and an eight-channel receive coil. Preamplifiers were located behind the animals, and the receive coil was placed inside an in-house built quadrature birdcage coil (12 cm inner diameter) used for transmission. Functional images were acquired during 6 functional runs for each animal using gradient-echo based single-shot echo-planar images (EPI) sequence with the following parameters: TR = 1.5 s, TE = 15ms, flip angle = 40°, field of view = 64 × 48 mm, matrix size = 96 × 128, resolution of 0.5 mm3 isotropic, number of slices = 42 [axial], bandwidth = 400 kHz, GRAPPA acceleration factor: 2 (left-right). Another set of EPIs with an opposite phase-encoding direction (right-left) was collected for the EPI-distortion correction. A T2-weighted structural was also acquired for each animal during one of the sessions with the following parameters: TR = 7 s, TE = 52ms, field of view = 51.2 × 51.2 mm, resolution of 0.133x0.133 × 0.5 mm, number of slices = 45 [axial], bandwidth = 50 kHz, GRAPPA acceleration factor: 2.

For human subjects, fMRI data were acquired on a 7T 68 cm MRI scanner (Siemens Magnetom 7T MRI Plus) with an AC-84 Mark II gradient coil, an in-house 8-channel parallel transmit, and a 32-channel receive coil (*Gilbert et al., 2021*). Functional images were acquired during 3 functional runs for each participant using Multi-Band EPI BOLD sequences with the following parameters: TR = 1.5 s, TE = 20ms, flip angle = 30°, field of view = 208 × 208 mm, matrix size = 104 × 104, resolution of 2 mm3 isotropic, number of slices = 62, GRAPPA acceleration factor: 3 (anterior-posterior), multi-band acceleration factor: 2. Field map images were also computed from the magnitude image and the two phase images. An MP2RAGE structural image was also acquired for each subject during the sessions with the following parameters: TR = 6 s, TE = 2.13ms, TI1 /TI2=800 / 2700ms, field of view = 240 × 240 mm, matrix size = 320 × 320, resolution of 0.75 mm3 isotropic, number of slices = 45, GRAPPA acceleration factor (anterior posterior): 3.

## MRI data preprocessing

Marmoset fMRI data were preprocessed using AFNI (*Cox, 1996*) and FSL (*Smith et al., 2004*) software packages. Raw MRI images were first converted to NIfTI format using dcm2nixx AFNI's function and then reoriented to the sphinx position using fslswapdim and fslorient FSL's functions. Functional images were despiked using 3Ddespike AFNI's function and time shifted using 3dTshift AFNI's function. Then, the images obtained were registered to the base volume (i.e., corresponding to the middle volume of each time series) with 3dvolreg AFNI's function. The output motion parameters obtained from volume registration were later used as nuisance regressors. All fMRI images were spatially smoothed with a 1.5 mm half-maximum Gaussian kernel (FWHM) with 3dmerge AFNI's function, followed by temporal filtering (0.01–0.1 Hz) using 3dBandpass AFNI's function. The mean functional image was calculated for each run and linearly registered to the respective anatomical image of each animal using FMRIB's linear registration tool (FLIRT).

The transformation matrix obtained after the registration was then used to transform the 4D time series data. The brain was manually skull-stripped from individual anatomical images using FSL eyes tool and the mask of each animal was applied to the functional images. Finally, the individual anatomical images were linearly registered to the NIH marmoset brain template (*Liu et al., 2018*) using Advanced Normalization Tools (ANTs).

Human fMRI data were preprocessed using SPM12 (Wellcome Department of Cognitive Neurology). After converting raw images into NifTI format, functional images were realigned to correct for head movements and underwent slice timing correction. A field map correction was applied to the functional images from the magnitude and phase images with the specify toolbox implemented in SPM. Then, the anatomical and functional volumes corrected were coregistered with the MP2RAGE structural scan from each individual participant and normalized to the Montreal Neurological Institute (MNI) standard brain space. Anatomical images were segmented into white matter, gray matter, and CSF partitions and also normalized to the MNI space. The functional images were then spatially smoothed with a 6 mm FWHM isotropic Gaussian kernel. A high-pass filter (128 s) was also applied to the time series.

## Statistical analysis

### Behavioral eye tracking data

To evaluate gaze patterns during observation of ToM and Random videos, we used mixed analyses of variance (ANOVA), with factors of species (Human vs Marmoset) and condition (ToM vs Random videos) on the overall fixation duration and on the proportion of time when the radial distance between the subject's gaze position and each triangle was less than 4 degrees. Partial eta squared ($\eta_p^2$) was computed as a measure of effect size and *post-hoc* comparisons were Bonferroni corrected.

### fMRI data

For each run, a general linear regression model was defined: the task timing was convolved to the hemodynamic response (AFNI's 'BLOCK' convolution for marmosets' data and SPM12 hemodynamic response function for humans' data) and a regressor was generated for each condition (AFNI's 3dDeconvolve function for marmosets and SPM12 function for humans). The two conditions were entered into the same model, corresponding to the 19.5 s presentation of the stimuli, along with polynomial detrending regressors and the marmosets' motions parameters or human's head movement parameters estimated during realignment.

The resultant regression coefficient maps of marmosets were then registered to template space using the transformation matrices obtained with the registration of anatomical images on the template (see MRI data processing part above).

Finally, we obtained for each run in marmosets and humans, two T-value maps registered to the NIH marmoset brain atlas (*Liu et al., 2018*) and to the MNI brain standard space, respectively.

These maps were then compared at the group level via paired t-tests using AFNI's 3dttest ++function, resulting in Z-value maps. To protect against false positives and to control for multiple comparisons, we adopted a clustering method derived from 10000 Monte Carlo simulations to the resultant

z-test maps using ClustSim option ($\alpha$=0.05). This method corresponds to performing cluster-forming threshold of p<0.01 uncorrected and then applying a family-wise error (FWE) correction of p<0.05 at the cluster-level.

We used the Paxinos parcellation of the NIH marmoset brain atlas (*Liu et al., 2018*) and the most recent multi-modal cortical parcellation atlas (*Glasser et al., 2016*) to define anatomical locations of cortical and subcortical regions for both marmosets and humans respectively.

First, we identified brain regions involved in the processing of ToM and Random animations by contrasting each condition with a baseline (i.e. ToM condition >baseline and Random condition >baseline contrasts). This baseline brain activation recorded during the presentation of the circular black cue between video clips (i.e. baseline blocks of 15 s, see above), reflects 'resting state' activation. By comparing it to the brain activation during ToM and Random animations, we could specifically highlight the task-related activations and isolate brain regions engaged during each condition. Subsequently, we then determined the clusters that displayed significantly greater activation for the ToM animations compared to the Random animations (ToM condition >Random condition contrast), and vice versa. The resultant Z-value maps were displayed on fiducial maps obtained from the Connectome Workbench (v1.5.0 [*Marcus et al., 2011*]) using the NIH marmoset brain template (*Liu et al., 2018*) for marmosets and the MNI Glasser brain template (*Glasser et al., 2016*) for humans. Subcortical activations were displayed on coronal sections.

As we used shortened video clips (i.e. 19.5 s compared to the 40 s originally designed by *Abell et al., 2000*), we validated our fMRI protocol by confirming that our shorter videos elicited similar responses to those previously observed in the HCP (*Barch et al., 2013*), whichalso used modified versions of these animation videos. We compared our ToM vs Random Z-value map obtained in human subjects with those of the HCP (*Barch et al., 2013*). To this end, we downloaded the Z-value map of activations for ToM animations compared to Random animations from 496 subjects from the Neurovalt site (https://identifiers.org/neurovault.image:3179). We displayed the resultant Z-value maps on fiducial maps obtained from the Connectome Workbench (v1.5.0, [*Marcus et al., 2011*]) using the MNI Glasser brain template (*Glasser et al., 2016*).

## Acknowledgements

Support was provided by the Canadian Institutes of Health Research (FRN 148365), the Canada First Research Excellence Fund to BrainsCAN, and a Discovery grant by the Natural Sciences and Engineering Research Council of Canada. We are grateful to Drs. Sarah White and Uta Frith for access to the Frith-Happé animation videos. We also wish to thank Cheryl Vander Tuin, Whitney Froese, Hannah Pettypiece, and Miranda Bellyou for animal preparation and care, Dr. Alex Li and Trevor Szekeres for scanning assistance, Dr. Kyle Gilbert and Peter Zeman for coil designs.

## Additional information

### Funding

| Funder | Grant reference number | Author |
| --- | --- | --- |
| Canadian Institutes of Health Research | FRN 148365 | Stefan Everling |
| Canada First Research Excellence Fund | | Stefan Everling |
| Natural Sciences and Engineering Research Council of Canada | Discovery grant | Stefan Everling |

The funders had no role in study design, data collection and interpretation, or the decision to submit the work for publication.

### Author contributions

Audrey Dureux, Conceptualization, Data curation, Formal analysis, Investigation, Methodology, Writing - original draft, Writing - review and editing; Alessandro Zanini, Conceptualization, Formal

analysis, Investigation, Methodology, Writing - review and editing; Janahan Selvanayagam, Formal analysis, Methodology, Writing - review and editing; Ravi S Menon, Project administration, Writing - review and editing; Stefan Everling, Conceptualization, Supervision, Funding acquisition, Investigation, Project administration, Writing - review and editing

### Author ORCIDs
Audrey Dureux (iD) http://orcid.org/0000-0003-1687-8347
Janahan Selvanayagam (iD) http://orcid.org/0000-0002-3708-8742
Stefan Everling (iD) http://orcid.org/0000-0001-9714-9757

### Ethics
Human subjects: This study was approved by the Ethics Committee of the University of Western Ontario and subjects were informed about the experimental procedures and provided informed written consent.

All experimental methods described were performed in accordance with the guidelines of the Canadian Council of Animal Care policy and a protocol approved by the Animal Care Committee of the University of Western Ontario Council on Animal Care (#2021-111). Animals were monitoring during the acquisition sessions by a veterinary technician.

### Decision letter and Author response
Decision letter https://doi.org/10.7554/eLife.86327.sa1
Author response https://doi.org/10.7554/eLife.86327.sa2

## Additional files

### Supplementary files
• MDAR checklist

### Data availability
All fMRI and eye tracking data generated and analysed as well as the scripts used have been deposited in Github and the link has been provided in the manuscript. Here the link: https://github.com/audreydureux/Theory-of-mind_Human_Marmosets_Paper (copy archived at *Dureux, 2023*).

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
