## [Editor Report]

Dureux and colleagues provide important evidence regarding the capacity for mental state attribution in a highly social non-human primate species, the marmoset. Their findings suggest that marmosets and humans visually track abstract stimuli more closely during ToM animations and display differential activation of large-scale networks implicated in social processing. These findings will be of wide interest to scientists interested in social cognition.

---

## [Decision Letter]

**Decision letter after peer review:**

Thank you for submitting your article "Gaze patterns and brain activations in humans and marmosets in the Frith-Happé theory-of-mind animation task" for consideration by *eLife*. Your article has been reviewed by 3 peer reviewers, one of whom is a member of our Board of Reviewing Editors, and the evaluation has been overseen by Timothy Behrens as the Senior Editor. The reviewers have opted to remain anonymous.

Essential revisions:

1. In the abstract, the claim that there is no evidence that nonhuman primates attribute mental states to moving shapes is false. You even cite some of this positive evidence (e.g., Uller, 2004; Atsumi et al., 2015; 2017). There is also evidence that they don't (Kupferberg et al., 2013; Burkart et al., 2012; Schafroth et al., 2021). The abstract would be stronger if written to represent the state of the field more accurately.

2. The overall conclusion as stated in the abstract, at the end of the introduction, and in the discussion is not warranted by the evidence. Indeed, the abstract completely fails to mention that the marmosets failed to show the human-like pattern of longer fixations on the ToM videos. Many readers will likely interpret this evidence as primarily against the idea that marmosets view the ToM videos in a human-like way, or as equivocal evidence at best. This report will be a stronger piece of science if it accurately describes the results.

3. The authors need to explicitly mention the rationale for omitting the original Goal-Directed condition from the Frith-Happé task. We cannot necessarily conclude that the marmosets are engaged in mental state attribution on the basis of these brain activation patterns – it could reflect the processing of distinct biological movements or the unfolding of an event narrative. If the authors proceed in pushing this data without the Goal-Directed videos, they must address their rationale for not testing these videos.

4. Were there any differences between the different types of videos used? For instance, was there any difference between videos in which the interaction between the two shapes is more obvious (e.g. coaxing vs seducing)?

5. Did the authors also present social videos to their animals? If so did they also observe additional recruitment of IPa and TPO areas like Clery and colleagues for social videos compared to Frith and Happe's videos (Cléry et al., 2021)?

6. Humans and marmosets recruit a distinct set of subcortical structures during the viewing of video clips; for instance dorsal thalamus and cerebellum in humans, the hippocampus and amygdala in marmosets. How do the authors interpret this difference?

7. Unlike marmosets, rhesus macaques are not sensitive to the type of the Frith and Happe social illusion (Roumazeilles et al., 2021; Schafroth et al., 2021). The authors might want to discuss the singularity of the marmosets from an evolutionary perspective.

8. The justification for looking in marmosets could be read to imply that macaque monkeys do not live in family groups or share important social similarities with humans. Both species share many social similarities (and many social differences) with humans. Marmosets are a good species to study; this section would benefit from a more accurate rationale.

9. Because it is one of the main metrics in the Klein and Schafroth papers, and thus readers will want to see it for sake of comparison, the authors should include a figure showing the overall fixation durations as a function of category and species.

10. The results about looking time to the large triangle need to follow up on the interaction between species and conditions so that readers know how to interpret it.

11. Are the bars in Figure 2 meant to add up to 1 for any given participant? If you analyzed the total time fixating on either shape, would marmosets be spending less time looking at the shapes overall than humans?

12. Readers will likely want clarification in cases where the same area showed stronger activation for ToM videos AND Random videos. I assume it was in different voxels in the same larger area, but this could be explicit.

13. The claim that these maps represent "dedicated brain networks" for ToM or Random videos (line 188) is too strong. These brain areas are used for many things.

14. For many of the sentences in the imaging results, the comparison needs to be made explicit. For example Line 193 – higher bilateral activation than what? Line 196 – greater activations than what? Line 202 – a larger network than what? Etc.

15. The description of Klein et al., (2009) on Lines 289-293 might be read to imply that they were attributing mentalizing without good reason. Klein also collected intentionality scores, which correlated with the viewing metric. This could be rephrased to be more accurate.

16. The inclusion of the authors as subjects is odd. Some readers will view it as a big red flag. The authors clearly know their own hypothesis and likely have a vested interest in a particular outcome. For the strongest report, the authors should remove their own data. At the very least, the authors need to demonstrate that the inclusion/exclusion of their unblinded data doesn't affect the interpretation of the human results.

17. The method should state whether the subjects had experienced these animations before (e.g., they're shown in some psychology and neuroscience classes).

18. The description of the monkey reward contingencies needs to be clearer about whether the monkeys were rewarded only during calibration or during videos as well, and whether any reward during videos was contingent on keeping their eyes on the screen.

19. Because this is a social task when the scans were normalized to MNI space, did the authors divide the human participants into those with and without a paracingulate sulcus?

20. The authors need to better specify what counts as a "baseline" for the fMRI comparisons. They should also briefly justify why this is an informative comparison.

*Reviewer #1 (Recommendations for the authors):*

I very much enjoyed reading this manuscript and believe it has the potential to make an important contribution to animal literature as well as social cognition more broadly.

As highlighted in the public review, my main query is in relation to the omission of the Goal-Directed condition of the Frith-Happé task. While the evidence presented here certainly suggests that marmosets process ToM animations in a different manner than Random animations, we are somewhat constrained in what we can interpret from these findings. As the authors note, we cannot necessarily conclude that the marmosets are engaged in mental state attribution on the basis of these brain activation patterns.

A more compelling argument would stem from the inclusion of the Goal-Directed condition in which the triangles arguably do interact but in a purely physical manner, i.e., there is no mental state attribution. I was surprised that this condition was not included as its omission somewhat limits the extent to which any conclusion regarding ToM can be drawn. Could the activations observed in the ToM condition reflect the processing of an event or narrative as it unfolds, rather than the cycling of random movements in the Random condition? I ask this question as previous studies using the Frith-Happé animations in dementia populations note that the mental state attribution judgements on ToM trials were conferred only at the end of the video (i.e., once the overall event narrative had been seen) whereas patients were adept at conferring a judgment of "no interaction" early during the viewing of Random animations (Ref: Synn et al. 2018 J. Alz Dis).

I wonder whether this interpretation might also reflect the curious finding of stronger medial PFC activation in Random trials versus ToM trials in humans, and no clear mPFC activation in the ToM trials. This seems very much at odds with the wider literature on the brain regions necessary for ToM, which often place the medial PFC at the heart of the social brain.

I very much appreciated the check using the independent HCP dataset. This was a very nice inclusion to ensure that the shortened version corresponded well with previous reports.

*Reviewer #2 (Recommendations for the authors):*

In their study, Dureux and colleagues are investigating the sensitivity of a highly social non-human primate species, the marmoset, to social illusion using the Frith and Happe task. Although this task is often considered a non-verbal TOM task, its relevance to investigate TOM has been disputed. For instance, the Frith and Happe task does not recruit in humans a similar network as other false-belief tasks and social gambling tasks (Schurz et al., 2020). While the authors might want to revise, or at least discuss their use of the TOM concept further, their results clearly show that marmosets distinguish the two types of videos shown to them.

Were there any differences between the different types of videos used? For instance, was there any difference between videos in which the interaction between the two shapes is more obvious (e.g. coaxing vs seducing).

Did the authors also present social videos to their animals? If so did they also observe additional recruitment of IPa and TPO areas like Clery and colleagues for social videos compared to Frith and Happe's videos (Cléry et al., 2021)?

Humans and marmosets recruit a distinct set of subcortical structures during the viewing of video clips; for instance dorsal thalamus and cerebellum in humans, and the hippocampus and amygdala in marmosets. How do the authors interpret this difference?

Unlike marmosets, rhesus macaques are not sensitive to the type of the Frith and Happe social illusion (Roumazeilles et al., 2021; Schafroth et al., 2021). The authors might want to discuss the singularity of the marmosets from an evolutionary perspective.

Refs:

Cléry JC, Hori Y, Schaeffer DJ, Menon RS, Everling S. 2021. Neural network of social interaction observation in marmosets. *eLife* 10:e65012. doi:10.7554/*eLife*.65012

Roumazeilles L, Schurz M, Lojkiewiez M, Verhagen L, Schüffelgen U, Marche K, Mahmoodi A, Emberton A, Simpson K, Joly O, Khamassi M, Rushworth MFS, Mars RB, Sallet J. 2021. Social prediction modulates activity of macaque superior temporal cortex (preprint). Neuroscience. doi:10.1101/2021.01.22.427803

Schafroth JL, Basile BM, Martin A, Murray EA. 2021. No evidence that monkeys attribute mental states to animated shapes in the Heider-Simmel videos. Sci Rep 11:3050. doi:10.1038/s41598-021-82702-6

Schurz M, Radua J, Tholen MG, Maliske L, Margulies DS, Mars RB, Sallet J, Kanske P. 2020. Toward a hierarchical model of social cognition: A neuroimaging meta-analysis and integrative review of empathy and theory of mind. Psychological Bulletin. doi:10.1037/bul0000303

*Reviewer #3 (Recommendations for the authors):*

This study is strong in many ways, and the goal is a good one. The below recommendations will help strengthen it further:

In the abstract, the claim that there is no evidence that nonhuman primates attribute mental states to moving shapes is false. You even cite some of this positive evidence (e.g., Uller, 2004; Atsumi et al., 2015; 2017). There is also evidence that they don't (Kupferberg et al., 2013; Burkart et al., 2012; Schafroth et al., 2021). The abstract would be stronger if written to represent the state of the field more accurately.

The overall conclusion as stated in the abstract, at the end of the introduction, and in the discussion is not warranted by the evidence. Indeed, the abstract completely fails to mention that the marmosets failed to show the human-like pattern of longer fixations on the ToM videos. Many readers will likely interpret this evidence as primarily against the idea that marmosets view the ToM videos in a human-like way, or as equivocal evidence at best. This report will be a stronger piece of science if it accurately describes the results.

The justification for looking in marmosets could be read to imply that macaque monkeys do not live in family groups or share important social similarities with humans. Both species share many social similarities (and many social differences) with humans. Marmosets are a good species to study; this section would benefit from a more accurate rationale.

Because it is one of the main metrics in the Klein and Schafroth papers, and thus readers will want to see it for sake of comparison, the authors should include a figure showing the overall fixation durations as a function of category and species.

The results about looking time to the large triangle need to follow up on the interaction between species and conditions so that readers know how to interpret it.

The sentence on lines 97-99 might be an incomplete sentence.

Are the bars in Figure 2 meant to add up to 1 for any given participant? If you analyzed the total time fixating on either shape, would marmosets be spending less time looking at the shapes overall than humans?

Overall, the figures are quite informative and aesthetically pleasing.

HCP should be explained the first time it is used.

Readers will likely want clarification in cases where the same area showed stronger activation for ToM videos AND Random videos. I assume it was in different voxels in the same larger area, but this could be explicit.

The claim that these maps represent "dedicated brain networks" for ToM or Random videos (line 188) is too strong. These brain areas are used for many things.

For many of the sentences in the imaging results, the comparison needs to be made explicit. For example Line 193 – higher bilateral activation than what? Line 196 – greater activations than what? Line 202 – a larger network than what? Etc.

The description of Klein et al., (2009) on Lines 289-293 might be read to imply that they were attributing mentalizing without good reason. Klein also collected intentionality scores, which correlated with the viewing metric. This could be rephrased to be more accurate.

In general, the discussion could be strengthened by avoiding repeating the results in as much detail.

The inclusion of the authors as subjects is odd. Some readers will view it as a big red flag. The authors clearly know their own hypothesis and likely have a vested interest in a particular outcome. For the strongest report, the authors should remove their own data. At the very least, the authors need to demonstrate that the inclusion/exclusion of their unblinded data doesn't affect the interpretation of the human results.

The method should state whether the subjects had experienced these animations before (e.g., they're shown in some psychology and neuroscience classes).

If the authors proceed in pushing this data without the Goal-Directed videos, they need to at least address their rationale for not testing these videos.

The description of the monkey reward contingencies needs to be clearer about whether the monkeys were rewarded only during calibration or during videos as well, and whether any reward during videos was contingent on keeping their eyes on the screen.

Because this is a social task when the scans were normalized to MNI space, did the authors divide the human participants into those with and without a paracingulate sulcus?

The authors need to better specify what counts as a "baseline" for the fMRI comparisons. They should also briefly justify why this is an informative comparison.

---

## [Author Response]

Essential revisions:1. In the abstract, the claim that there is no evidence that nonhuman primates attribute mental states to moving shapes is false. You even cite some of this positive evidence (e.g., Uller, 2004; Atsumi et al., 2015; 2017). There is also evidence that they don't (Kupferberg et al., 2013; Burkart et al., 2012; Schafroth et al., 2021). The abstract would be stronger if written to represent the state of the field more accurately.

We greatly appreciate the reviewer's insightful comments. In response, we have revised the abstract to better align with the existing literature concerning nonhuman primates’ abilities to attribute mental states to moving shapes. The specific modifications can be found on page 2 of the revised manuscript.

Page 2: “Theory of Mind (ToM) refers to the cognitive ability to attribute mental states to other individuals. This ability extends even to the attribution of mental states to animations featuring simple geometric shapes, such as the Frith-Happé animations in which two triangles move either purposelessly (Random condition), exhibit purely physical movement (Goal-directed condition), or move as if one triangle is reacting to the other triangle’s mental states (ToM condition). While this capacity in humans has been thoroughly established, research on nonhuman primates has yielded inconsistent results.

This study explored how marmosets (Callithrix jacchus), a highly social primate species, process Frith-Happé animations by examining gaze patterns and brain activations of marmosets and humans as they observed these animations. We revealed that both marmosets and humans exhibited longer fixations on one of the triangles in ToM animations, compared to other conditions. However, we did not observe the same pattern of longer overall fixation duration on the ToM animations in marmosets as identified in humans. Furthermore, our findings reveal that both species activated extensive and comparable brain networks when viewing ToM versus Random animations, suggesting that marmosets differentiate between these scenarios similarly to humans. While marmosets did not mimic human overall fixation patterns, their gaze behavior and neural activations indicate a distinction between ToM and non-ToM scenarios. This study expands our understanding of nonhuman primate cognitive abilities, shedding light on potential similarities and differences in ToM processing between marmosets and humans.”

2. The overall conclusion as stated in the abstract, at the end of the introduction, and in the discussion is not warranted by the evidence. Indeed, the abstract completely fails to mention that the marmosets failed to show the human-like pattern of longer fixations on the ToM videos. Many readers will likely interpret this evidence as primarily against the idea that marmosets view the ToM videos in a human-like way, or as equivocal evidence at best. This report will be a stronger piece of science if it accurately describes the results.

We agree that it is crucial to precisely represent the results of our study, including the nuanced details about marmosets' reactions to the ToM videos. To this end, we have revised the abstract, introduction, and Discussion sections to provide a more balanced and precise interpretation of our findings.

The major revisions appear on page 2 in the abstract, on page 4 in the introduction, and between pages 14 to 20 in the Discussion section. We believe that these changes will ensure the results and conclusions of the study are conveyed more accurately and transparently.

The revised content reads as follows:

Page 2: “Theory of Mind (ToM) refers to the cognitive ability to attribute mental states to other individuals. This ability extends even to the attribution of mental states to animations featuring simple geometric shapes, such as the Frith-Happé animations in which two triangles move either purposelessly (Random condition), exhibit purely physical movement (Goal-directed condition), or move as if one triangle is reacting to the other triangle’s mental states (ToM condition). While this capacity in humans has been thoroughly established, research on nonhuman primates has yielded inconsistent results.

This study explored how marmosets (Callithrix jacchus), a highly social primate species, process Frith-Happé animations by examining gaze patterns and brain activations of marmosets and humans as they observed these animations. We revealed that both marmosets and humans exhibited longer fixations on one of the triangles in ToM animations, compared to other conditions. However, we did not observe the same pattern of longer overall fixation duration on the ToM animations in marmosets as identified in humans. Furthermore, our findings reveal that both species activated extensive and comparable brain networks when viewing ToM versus Random animations, suggesting that marmosets differentiate between these scenarios similarly to humans. While marmosets did not mimic human overall fixation patterns, their gaze behavior and neural activations indicate a distinction between ToM and non-ToM scenarios. This study expands our understanding of nonhuman primate cognitive abilities, shedding light on potential similarities and differences in ToM processing between marmosets and humans.”

Pages 4-5: “Although the spontaneous attribution of mental states to moving shapes has been well established in humans, it remains uncertain whether other primate species share this capacity. There is some evidence suggesting that monkeys can attribute goals to agents with varying levels of similarity and familiarity to conspecifics, including human agents, monkey robots, moving geometric boxes, animated shapes, and simple moving dots (Atsumi et al., 2017; Atsumi and Nagasaka, 2015; Krupenye and Hare, 2018; Kupferberg et al., 2013; Uller, 2004). However, the findings in this area are somewhat mixed, with some studies investigating the attribution of goals to inanimate moving objects yielding inconclusive results (Atsumi and Nagasaka, 2015; Kupferberg et al., 2013). Nonhuman primates' spontaneous attribution of mental states to Frith-Happé animations is even less certain. While human subjects exhibit longer eye fixations when viewing the ToM condition compared to the Random condition of the Frith-Happé animations (Klein et al., 2009), a recent eye tracking study in macaque monkeys did not observe similar differences (Schafroth et al., 2021). Similarly, a recent fMRI study conducted on macaques found no discernible differences in activations between ToM and random Frith-Happé animations (Roumazeilles et al., 2021).

In this study, we investigated the behaviour and brain activations of New World common marmoset monkeys (Callithrix jacchus) while they viewed Frith-Happé animations. Living in closely-knit family groups, marmosets exhibit significant social parallels with humans, including prosocial behavior, imitation, and cooperative breeding. These characteristics establish them as a promising nonhuman primate model for investigating social cognition (Burkart et al., 2009; Burkart and Finkenwirth, 2015; Miller et al., 2016). To directly compare humans and marmosets in their response to these animations, we employed high-speed video eye-tracking to record eye movements in eleven healthy humans and eleven marmoset monkeys. Additionally, we conducted ultra-high field fMRI scans on ten healthy humans at 7T and six common marmoset monkeys at 9.4T. These combined methods allowed us to examine the visual behavior and brain activations of both species while they observed the Frith-Happé animations.”

Pages 14-15: “In our first experiment, we examined the gaze patterns of marmosets and humans during the viewing of these video animations. Klein et al. (2009) reported differing fixation durations for these animations, where the longest fixations were observed for ToM animations, followed by GD animations and the shortest fixations for Random animations. They further reported that the intentionality score – derived from verbal descriptions of the animations – followed a similar pattern: highest for ToM, lowest for Random, and intermediate for GD animations. This validated the degree of mental state attribution according to the categories and established that animations provoking mentalizing (ToM condition) were associated with long fixations. This, in turn, supports the use of fixation durations as a nonverbal metric for mentalizing capacity (Klein et al., 2009; Meijering et al., 2012). Our results with human subjects, which demonstrated longer fixation durations for the ToM animations compared to the GD and Random animations, paralleled those of Klein et al. (2009). However, unlike Klein et al.'s findings, we did not observe intermediate durations for GD animations in our study.

Interestingly, our marmoset data did not align with the human findings but instead resonated more with Schafroth et al. (2021)'s observations in macaque monkeys, which did not show significant differences in fixation durations across the three animation types.”

Pages 19-20: “In summary, our study reveals novel insights into how New World marmosets, akin to humans, differentially process abstract animations that depict complex social interactions and animations that display purely physical or random movements. Our findings, supported by both specific gaze behaviors (i.e., the proportion of time spent on the red triangle, despite the inconclusiveness of overall fixation) and distinct neural activation patterns, shed light on the marmosets' capacity to interpret social cues embedded in these animations.

The differences observed between humans, marmosets, and macaques underscore the diverse cognitive strategies that primate species have evolved to decipher social information. This diversity may be influenced by unique evolutionary pressures that arise from varying social structures and lifestyles. Like macaque monkeys, humans often live in large, hierarchically organized social groups where status influences access to resources. However, both humans and marmosets share a common trait: a high degree of cooperative care for offspring within the group, with individuals other than the biological parents participating in child-rearing. These distinctive social dynamics of marmosets and humans may have driven the development of unique social cognitive abilities. This could explain their enhanced sensitivity to abstract social cues in the Frith-Happé animations.

Nonetheless, it is crucial to emphasize that even though marmosets respond to the social cues in the Frith-Happé animations, this does not automatically imply that they possess mental-state attributions comparable to humans. As such, future research including a range of tasks, from sensory-affective components to more abstract and decoupled representations of others' mental states (Schurz et al., 2020), will be fundamental in further unravelling the complexities of the evolution and functioning of the theory of mind across the primate lineage.”

3. The authors need to explicitly mention the rationale for omitting the original Goal-Directed condition from the Frith-Happé task. We cannot necessarily conclude that the marmosets are engaged in mental state attribution on the basis of these brain activation patterns – it could reflect the processing of distinct biological movements or the unfolding of an event narrative. If the authors proceed in pushing this data without the Goal-Directed videos, they must address their rationale for not testing these videos.

We appreciate the reviewer's feedback about the omission of the Goal-Directed condition from the Frith-Happé task in our study. In our revised manuscript, we have elaborated on the factors influencing our decision to focus primarily on the ToM and Random conditions. These factors were two-fold:

1. Influence from prior fMRI studies: Many previous fMRI studies using the Frith-Happé animated triangles task with human and macaque subjects have only employed the ToM and Random conditions. These conditions represent the two extremes, with ToM depicting scenarios with mental interactions and Random showing scenarios absent of mental interactions. GD condition is situated between these two extremes, depicting physical interaction among the triangles without suggesting mental state attribution.

2. Practical considerations: The duration of each video clip in the Frith-Happé task (19.5 seconds) presented challenges for keeping marmoset subjects alert and focused during longer scanning sessions.

Given these constraints, we made the decision to limit the number of conditions presented in a single run.

However, understanding the value of including the Goal-Directed condition, we have performed an additional eye-tracking experiment incorporating all three conditions: ToM, Goal-Directed, and Random. The results from this experiment provided further insights into the gaze patterns during these conditions, adding depth to our understanding of marmoset behavior during the different conditions.

Moreover, while our data reveal distinct patterns of brain activation and gaze behavior during the ToM condition, we recognize and emphasize in our revised manuscript that these patterns do not conclusively prove that marmosets attribute mental states in the same way as humans.

These adjustments, along with the findings from the new eye-tracking experiment, have been integrated into the revised manuscript. The relevant sections in the methods, results, and discussion have been modified accordingly. These revisions can be found on pages 5 to 7 (Results section), 14 to 20 (Discussion section), and 23-24 (method section) of the revised manuscript.

4. Were there any differences between the different types of videos used? For instance, was there any difference between videos in which the interaction between the two shapes is more obvious (e.g. coaxing vs seducing)?

We appreciate the reviewer's interest in the distinct types of videos used in our study. In this investigation, we did not specifically analyze the responses to videos where the interaction between the two shapes was more or less pronounced, such as in coaxing versus seducing scenarios. Our primary focus was on contrasting the overall responses elicited by ToM animations and Random animations. Due to our experimental design, we did not have enough repetitions of each distinct type of video within each condition to provide the statistical power necessary for such an analysis.

We recognize that dissecting responses to different types of ToM animations might reveal further insights into the specificity of neural responses, and this is an intriguing area for future research.

5. Did the authors also present social videos to their animals? If so did they also observe additional recruitment of IPa and TPO areas like Clery and colleagues for social videos compared to Frith and Happe's videos (Cléry et al., 2021)?

Our study specifically focused on neural responses to the abstract social scenarios represented by the Frith-Happé animations. Consequently, realistic social videos were not included in our experimental design.

The study by Cléry et al. (2021) indeed demonstrates that the comparison social versus non-social realistic videos predominantly revealed a fronto-parietal network with additional temporal region engagement. The social condition in their study seems to recruit not only several areas that we observed to be activated in our ToM condition, but also additional temporal regions such as the IPa, TPO and TE1 areas. This suggests that these areas may play a significant role in the processing of more realistic social cues, thereby adding complexity to the social brain network.

Although our current study did not directly investigate this aspect, we agree that comparing the neural responses elicited by abstract versus realistic social scenarios in marmosets could provide valuable insights into the extent and adaptability of the social brain network. This approach would further our understanding of the neural substrates that underpin various facets of social cognition, depending on the complexity and realism of the presented stimuli. In future research, we plan to consider including such comparative investigations in our experimental design. We appreciate the reviewer's suggestion.

6. Humans and marmosets recruit a distinct set of subcortical structures during the viewing of video clips; for instance dorsal thalamus and cerebellum in humans, the hippocampus and amygdala in marmosets. How do the authors interpret this difference?

We are grateful to the reviewer for drawing attention to the distinct set of subcortical structures engaged by humans and marmosets during the viewing of video clips. The divergent patterns may reflect species-specific social cognitive strategies.

In humans, the involvement of the dorsal thalamus, which serves as a critical hub for information relay between various subcortical areas and the cortex, may indicate the necessity for complex information processing in interpreting the animations (e.g., Halassa and Sherman, 2019). The activation of the cerebellum, beyond its traditional role in motor functions, supports recent findings of its involvement in social cognition (e.g., Van Overwalle et al., 2014). The activations in a small portion of the amygdala may reflect emotional processing tied to understanding of the social scenarios in the animations (e.g., Janak and Tye, 2015).

On the other hand, the activation of the hippocampus and amygdala in marmosets might reflect a more emotion-driven interpretation of the animations. The recruitment of the hippocampus could suggest the role of memory in interpreting the animations by remembering past interactions to help interpret current social scenarios (e.g., Eichenbaum, 2017), while the more extended activation of the amygdala in marmosets might imply a higher degree of emotional processing compared to humans (e.g., Janak and Tye, 2015).

In response, we have included a detailed discussion in our revised manuscript, which takes into account the potential roles of these structures in complex information processing, social cognition, emotional processing, and memory recall in the context of interpreting the animations. We emphasize that these interpretations remain speculative and underscore the need for further research to confirm these observations.

We have now expanded our discussion on this topic in the revised manuscript, specifically on pages 18-19:

" Regarding the distinct subcortical activations observed in humans and marmosets, it's important to consider the specific social cognitive demands that might be unique to each species. The involvement of the dorsal thalamus, cerebellum, and a small portion of the amygdala in humans may reflect the complexities of information processing, social cognition, and emotional involvement required to interpret the ToM animations (e.g., Halassa and Sherman, 2019; Janak and Tye, 2015; Van Overwalle et al., 2014). Conversely, the activation of the amygdala and hippocampus in marmosets could suggest a more emotion- and memory-based processing of the social stimuli (e.g., Eichenbaum, 2017; Van Overwalle et al., 2014). However, it's critical to consider that these interpretations are speculative and would require further study for confirmation.”

7. Unlike marmosets, rhesus macaques are not sensitive to the type of the Frith and Happe social illusion (Roumazeilles et al., 2021; Schafroth et al., 2021). The authors might want to discuss the singularity of the marmosets from an evolutionary perspective.

As suggested, in our revised manuscript, we have now incorporated a comprehensive discussion regarding the uniqueness of marmosets from an evolutionary perspective, especially in light of the different results obtained in a similar study conducted on rhesus macaques. We highlight the divergent evolutionary trajectories of New World monkeys (such as marmosets) and Old-World monkeys (such as macaques), which may contribute to the differential sensitivity to abstract social cues embedded in animations.

We also underscore the diverse cognitive strategies that primate species employ in deciphering social information, influenced by unique evolutionary pressures arising from varying social structures and lifestyles.

The associated changes in our manuscript can be found in the Discussion section on pages 19-20, which reads:

“Interestingly, our results differed from those obtained by Roumazeilles et al. (2021) in their fMRI study conducted in macaques using the same animations. Roumazeilles and colleagues reported no differences in activation between ToM and Random animations, suggesting that rhesus macaques may not respond to the social cues presented by the ToM Frith-Happé animations. This disparity between our marmoset findings and those of macaques raises intriguing questions about potential differences in the evolutionary development of ToM processing within non-human primates. Marmosets, as New World monkeys, are part of an evolutionary lineage that diverged earlier than the lineage of Old-World monkeys such as macaques. This difference in lineage might lead to distinct evolutionary trajectories in cognitive processing, which could include varying sensitivity to abstract social cues in animations.

In summary, our study reveals novel insights into how New World marmosets, akin to humans, differentially process abstract animations that depict complex social interactions and animations that display purely physical or random movements. Our findings, supported by both specific gaze behaviors (i.e., the proportion of time spent on the red triangle, despite the inconclusiveness of overall fixation) and distinct neural activation patterns, shed light on the marmosets' capacity to interpret social cues embedded in these animations.

The differences observed between humans, marmosets, and macaques underscore the diverse cognitive strategies that primate species have evolved to decipher social information. This diversity may be influenced by unique evolutionary pressures that arise from varying social structures and lifestyles. Like macaque monkeys, humans often live in large, hierarchically organized social groups where status influences access to resources. However, both humans and marmosets share a common trait: a high degree of cooperative care for offspring within the group, with individuals other than the biological parents participating in child-rearing. These distinctive social dynamics of marmosets and humans may have driven the development of unique social cognitive abilities. This could explain their enhanced sensitivity to abstract social cues in the Frith-Happé animations.

Nonetheless, it is crucial to emphasize that even though marmosets respond to the social cues in the Frith-Happé animations, this does not automatically imply that they possess mental-state attributions comparable to humans. As such, future research including a range of tasks, from sensory-affective components to more abstract and decoupled representations of others' mental states (Schurz et al., 2020), will be fundamental in further unravelling the complexities of the evolution and functioning of the theory of mind across the primate lineage.”

8. The justification for looking in marmosets could be read to imply that macaque monkeys do not live in family groups or share important social similarities with humans. Both species share many social similarities (and many social differences) with humans. Marmosets are a good species to study; this section would benefit from a more accurate rationale.

In response to the reviewer's comment, we have clarified our previous section. Our intention was to highlight the unique social aspects of marmosets that make them a suitable species for studying social cognition, not to imply that macaques do not have their own set of social similarities with humans. As suggested, we have now revised the sentence on page 4 to enhance its clarity and accuracy, which now reads:

Page 4: “Living in closely-knit family groups, marmosets exhibit significant social parallels with humans, including prosocial behavior, imitation, and cooperative breeding. These characteristics establish them as a promising nonhuman primate model for investigating social cognition (Burkart et al., 2009; Burkart and Finkenwirth, 2015; Miller et al., 2016).”

9. Because it is one of the main metrics in the Klein and Schafroth papers, and thus readers will want to see it for sake of comparison, the authors should include a figure showing the overall fixation durations as a function of category and species.

In accordance with the reviewer's recommendation, we have included a new figure (Figure 2) into our revised manuscript, which can be found on page 39. This figure graphically represents the overall fixation durations as a function of both animation category and species. This facilitates a more comprehensive comparison of fixation durations between humans and marmosets across the different animation conditions (Random, Goal-directed, and ToM). This comparison is in alignment with the data representation found in the Klein and Schafroth paper.

10. The results about looking time to the large triangle need to follow up on the interaction between species and conditions so that readers know how to interpret it.

In response to the reviewer's comment, we have provided a more detailed analysis of the interaction between species and conditions for the proportion of time spent looking at the large red triangle. We found that both humans and marmosets spent a greater proportion of time looking at the red triangle in the ToM condition compared to the GD and Random conditions. However, while humans also allocated more time to the red triangle in GD compared to Random animations, marmosets did not show any difference between these two conditions. The updated text can be found in the Results section on pages 5 to 7, which now reads as follows:

Pages 6-7: “To further analyze the gaze patterns of both humans and marmosets, we next measured the proportion of time subjects looked at each of the triangles in the videos (Figure 2B). We conducted mixed ANOVAs on the proportion of time the radial distance between the current gaze position and each triangle was within 4 visual degrees for each triangle separately.

Importantly, we observed a significant interaction between species and condition for the proportion of time spent looking at the large red triangle (F(2,40)=9.83, p<.001, ηp2 = .330). Specifically, both humans (Figure 2B left) and marmosets (Figure 2B right) spent a greater proportion of time looking at the red triangle in ToM compared to the GD and Random videos (For humans, ToM vs GD: Δ=.23, p<.001 and ToM vs Random: Δ=.31, p<.001 ; For marmosets, ToM vs GD: Δ=.13, p<.01 and ToM vs Random: Δ=.13, p<.01). However, while humans also allocated a greater proportion of time to the red triangle in GD compared to Random animations (Δ=.08, p=.05), marmosets did not show any difference between these two conditions (Δ=.0003, p=1).

For the small blue triangle, we also observed a significant interaction of species and condition (F(2,40)=3.54, p=.04, ηp2=.151) but the comparisons were not resistant to the p value adjustment by Bonferroni correction. Therefore, humans and marmosets spent the same proportion of time looking at the blue triangle in the three different types of videos (For humans, ToM vs GD: Δ=-.02, p=1, ToM vs Random: Δ=.04, p=1 and GD vs Random: Δ=.07, p=.23 ; For marmosets, ToM vs GD: Δ=-.05, p=.89, ToM vs Random: Δ=.07, p=.66 and GD vs Random: Δ=-.02, p=1; Figure 2B).

These results highlight the variation in gaze patterns observed in both humans and marmosets when their focus is directed towards the large red triangle during the viewing of ToM, GD, and Random videos. Notably, humans show a gradient of proportion of time spent looking at the red triangle across the three conditions, with the smallest proportion in Random videos and the greatest proportion in ToM videos. In contrast, marmosets exhibit a different pattern, spending more time looking at the red triangle in ToM videos, but allocating the same proportion of time to look at the red triangle in both Random and GD videos. This finding suggests that while humans demonstrate distinct attentional preferences for the red triangle across the three conditions, marmosets exhibit a similar attentional focus on the red triangle in the Random and GD conditions, but their pattern differs in the ToM condition. This suggests that marmosets process the Random and GD conditions in a similar manner, but their processing of the ToM condition is distinct, indicating a differential response to stimuli representing social interactions.”

11. Are the bars in Figure 2 meant to add up to 1 for any given participant? If you analyzed the total time fixating on either shape, would marmosets be spending less time looking at the shapes overall than humans?

The values in the original Figure 2 do not total 1 for each participant, as there are instances where the triangles either overlap or are proximate enough that the eye position falls within the defined radius for both shapes simultaneously. Responding to the second query, after conducting an additional analysis, we found a significant species effect on the total time spent fixating on either shape (*F*_(1,20)_=14.38, *p*=.001, *η_p_^2^*=.42). This indicates that humans tend to look at the triangles more frequently than marmosets (Δ=.16, *p*=.001).

12. Readers will likely want clarification in cases where the same area showed stronger activation for ToM videos AND Random videos. I assume it was in different voxels in the same larger area, but this could be explicit.

In response to the reviewer's request for clarification, we have made it explicit in our manuscript that while larger areas of the brain showed stronger activation for both ToM and Random videos, the specific voxels within these areas exhibiting this activation were typically distinct. Furthermore, we note that in some instances, both conditions activated the same voxels, but the degree of activation differed, suggesting spatial and intensity variation within the same areas. This elaboration can be found in the sections discussing functional brain activations in humans (pages 7-8) and marmosets (page 11). This can be read:

Pages 7-8 (Functional brain activations while watching ToM and Random Frith-Happé’s animations in humans): “Both ToM (Figure 3A) and Random (Figure 3B) videos activated a large bilateral network. While the same larger areas were activated in both conditions, the specific voxels showing this activation within those areas were typically distinct. In some cases, both conditions activated the same voxels, but the degree of activation differed. This suggests a degree of both spatial and intensity variation in the activations for the two conditions within the same areas.”

Page 11 (Functional brain activations while watching ToM and Random Frith-Happé’s animations in marmosets): “Both the ToM (Figure 4A) and Random (Figure 4B) animations activated an extensive network involving a variety of areas in the occipito-temporal, parietal and frontal regions. As in human subjects, it should be noted that while both conditions elicited strong activation in some of the same larger areas, these activations might have either occurred in distinct voxels within those areas, or the same voxels were activated to varying degrees for both conditions. This suggests distinct yet overlapping patterns of neural processing for the ToM and Random conditions.”

13. The claim that these maps represent "dedicated brain networks" for ToM or Random videos (line 188) is too strong. These brain areas are used for many things.

In response to the reviewer's comment, we agree that the term "dedicated brain networks" could potentially imply exclusivity, which is not our intention. We understand that these brain areas participate in a variety of cognitive functions. To address this, we have modified our phrasing on page 10 to "brain networks activated during the processing of ToM or Random videos" to more accurately represent our findings.

14. For many of the sentences in the imaging results, the comparison needs to be made explicit. For example Line 193 – higher bilateral activation than what? Line 196 – greater activations than what? Line 202 – a larger network than what? Etc.

We agree with the reviewer's observation about the need for explicit comparisons in our imaging results. To address this, we have now revised certain sentences in the Results section to provide clear and specific comparisons. The updated descriptions can be found in the Results section on pages 7 to 12 of the revised manuscript.

15. The description of Klein et al., (2009) on Lines 289-293 might be read to imply that they were attributing mentalizing without good reason. Klein also collected intentionality scores, which correlated with the viewing metric. This could be rephrased to be more accurate.

We concur with the reviewer's suggestion for a more accurate interpretation of Klein et al., 2009. Our intention was not to undermine the work by Klein et al. To address this, we have adjusted the phrasing within the Discussion section on page 15 of our manuscript, emphasizing Klein et al.'s valuable contribution through their correlation of intentionality scores with fixation durations. These revisions result in a more balanced and accurate representation of their work.

Page 15: “In our first experiment, we examined the gaze patterns of marmosets and humans during the viewing of these video animations. Klein et al. (2009) reported differing fixation durations for these animations, where the longest fixations were observed for ToM animations, followed by GD animations and the shortest fixations for Random animations. They further reported that the intentionality score – derived from verbal descriptions of the animations – followed a similar pattern: highest for ToM, lowest for Random, and intermediate for GD animations. This validated the degree of mental state attribution according to the categories and established that animations provoking mentalizing (ToM condition) were associated with long fixations. This, in turn, supports the use of fixation durations as a nonverbal metric for mentalizing capacity (Klein et al., 2009; Meijering et al., 2012).”

16. The inclusion of the authors as subjects is odd. Some readers will view it as a big red flag. The authors clearly know their own hypothesis and likely have a vested interest in a particular outcome. For the strongest report, the authors should remove their own data. At the very least, the authors need to demonstrate that the inclusion/exclusion of their unblinded data doesn't affect the interpretation of the human results.

We acknowledge the reviewer's concern regarding the inclusion of authors as subjects, and potential bias it could introduce. In response to this, we have excluded the data from the authors who initially participated in the study and replaced it with new data from subjects unrelated to the authorship of this work. For the eye tracking experiment, given the introduction of a new “Goal-Directed condition”, we carried out the experiment with eleven new participants, none of whom are authors of this study. Regarding the fMRI experiment, we substituted the data collected from the three author-participants with data from three additional participants who were not informed about the study's hypothesis.

The re-analyzed results, accounting for the updated participant pool, can now be found in the sections: “Gaze patterns for Frith-Happé’s ToM, GD and Random animations in humans and marmosets” (pages 5-7), “Functional brain activations while watching ToM and Random Frith-Happé’s animations in humans” (pages 7-10), and “Comparison of functional brain activations in humans and marmosets” (pages 13-14). We also updated the figures 2, 3 and 5 and the figures supplement 1 and 2 on pages 39, 40, 42, 43, and 44, respectively.

The revised participant information is detailed in the methods section on page 22:

“Eleven healthy humans (4 females, 25-42 years, mean age: 30.7 years) participated in the eye tracking experiment. Among these, five individuals, along with eight additional subjects (4 females, 26-45 years), took part in the fMRI experiment.”

17. The method should state whether the subjects had experienced these animations before (e.g., they're shown in some psychology and neuroscience classes).

We agree that detailing whether subjects had previous exposure to the animations is essential for the study's integrity. As a result, we have incorporated the following statement into the Methods section on page 22:

Page 22: “Importantly, all subjects confirmed they had not previously been exposed to the Frith-Happé animation videos used in our study.”

18. The description of the monkey reward contingencies needs to be clearer about whether the monkeys were rewarded only during calibration or during videos as well, and whether any reward during videos was contingent on keeping their eyes on the screen.

We appreciate the reviewer's suggestion to provide additional clarification on the reward contingencies for the monkey in our study. The monkeys received rewards only at the initial and final stages of each session, but not during the calibration or the viewing of the videos. Consequently, we have updated the text in the Methods section, now found on page 23, as follows:

“Monkeys were rewarded at the beginning and end of each session. Crucially, no rewards were provided during the calibration or while the videos were played.”

19. Because this is a social task when the scans were normalized to MNI space, did the authors divide the human participants into those with and without a paracingulate sulcus?

We appreciate the reviewer's insightful comment. While we normalized our MRI scans to MNI space in this study, we did not differentiate among participants based on the presence or absence of a paracingulate sulcus. The reviewer’s suggestion to consider this factor into account in our analyses is indeed valuable and will be considered in our future studies involving a larger pool of participants.

20. The authors need to better specify what counts as a "baseline" for the fMRI comparisons. They should also briefly justify why this is an informative comparison.

We agree that the definition and justification for our selected "baseline" in the fMRI comparisons should be more explicit. In our study, the "baseline" denotes brain activity when subjects are in a 'resting state' – a state of neutral alertness – specifically during the presentation of a circular black cue between video clips. The comparison to this baseline is valuable because it allows us to isolate and constrast brain activity associated with task-specific conditions, such as ToM or Random animations. We have added a more detailed explanation in the Methods section on page 30 of the revised manuscript. It now reads:

“First, we identified brain regions involved in the processing of ToM and Random animations by contrasting each condition with a baseline (i.e., ToM condition > baseline and Random condition > baseline contrasts). This baseline brain activation recorded during the presentation of the circular black cue between video clips (i.e., baseline blocks of 15 sec, see above), reflects 'resting state' activation. By comparing it to the brain activation during ToM and Random animations, we could specifically highlight the task-related activations and isolate brain regions engaged during each condition.”

Reviewer #1 (Recommendations for the authors):I very much enjoyed reading this manuscript and believe it has the potential to make an important contribution to animal literature as well as social cognition more broadly.As highlighted in the public review, my main query is in relation to the omission of the Goal-Directed condition of the Frith-Happé task. While the evidence presented here certainly suggests that marmosets process ToM animations in a different manner than Random animations, we are somewhat constrained in what we can interpret from these findings. As the authors note, we cannot necessarily conclude that the marmosets are engaged in mental state attribution on the basis of these brain activation patterns.

We are grateful for the reviewer's encouraging words, thoughtful evaluation, and constructive comments on our manuscript. We agree with the reviewer's remarks regarding the omission of the Goal-Directed condition in the Frith-Happé task and recognize the interpretative constraints this places on our findings. We have attempted to address all points in both our responses below and in the revised manuscript.

A more compelling argument would stem from the inclusion of the Goal-Directed condition in which the triangles arguably do interact but in a purely physical manner, i.e., there is no mental state attribution. I was surprised that this condition was not included as its omission somewhat limits the extent to which any conclusion regarding ToM can be drawn. Could the activations observed in the ToM condition reflect the processing of an event or narrative as it unfolds, rather than the cycling of random movements in the Random condition? I ask this question as previous studies using the Frith-Happé animations in dementia populations note that the mental state attribution judgements on ToM trials were conferred only at the end of the video (i.e., once the overall event narrative had been seen) whereas patients were adept at conferring a judgment of "no interaction" early during the viewing of Random animations (Ref: Synn et al. 2018 J. Alz Dis).

We appreciate the reviewer's insightful comment concerning the absence of the Goal-Directed (GD) condition in our study. We understand that integrating this condition could have offered a valuable contrast and enriched our understanding of the associated processing mechanisms.

In our study, our initial strategy was to concentrate on the two extreme conditions: ToM and Random animations. These represent scenarios with mental interactions and scenarios with absence of mental interactions, respectively. This choice was influenced by some previous fMRI studies using the Frith-Happé animated triangles task in humans and macaques, which primarily focused on these two conditions (Gobbini et al., 2007; Barch et al., 2013; Bliksted et al., 2019; Vandewouw et al., 2021; Weiss et al., 2021; Chen et al., 2023; Roumazeilles et al., 2021). Additionally, the duration of each video clip (19.5 sec) posed practical challenges in incorporating all the conditions with a sufficient number of repetitions in the fMRI task design for marmoset subjects. It was crucial for us to ensure that the subjects remained alert and focused throughout the entire scanning session, which becomes increasingly difficult with longer runs. Consequently, we chose to center our attention on the ToM and Random conditions, as the GD condition is situated between these two extremes, depicting physical interaction among the triangles without suggesting mental state attribution.

Nevertheless, we recognize the potential limitations of not incorporating the GD condition and the possible insights it might offer. In response to the reviewer's feedback, we conducted an additional eye-tracking experiment that included all three conditions: ToM, GD, and Random. This experiment involved 11 human subjects and 11 marmosets, with all ToM, GD and Random video clips presented once in a single run. The results from this experiment provided additional insights into the gaze patterns during the different conditions, complementing our initial findings.

We have updated the manuscript to clarify the choice of two conditions for the fMRI experiment and to incorporate the findings from the new eye-tracking experiment. The relevant sections in the methods, results, and discussion have been modified accordingly. These revisions can be found on pages 23-24, 5 to 7, and 14 to 20, respectively.

The reviewer's insightful question regarding whether the observed activations in the ToM condition might simply reflect the processing of an event or narrative as it unfolds, rather than mental state attribution, is an important consideration. We understand from the current literature, including the Synn et al. (2018) study mentioned by the reviewer, that distinguishing between these two processes can be challenging, particularly given the dynamic nature of the stimuli used. The ToM condition intrinsically involves the progression of an event or narrative, which is necessary for subjects to infer the mental states of the characters.

However, we believe our results provide evidence of the specific involvement of certain brain regions in mental state attribution. The enhanced activation of certain brain regions (e.g., TPJ and STS) during the ToM condition compared to Random condition aligns with several prior fMRI studies (Barch et al., 2013; Castelli et al., 2000; Chen et al., 2023; Gobbini et al., 2007; Vandewouw et al., 2021; Weiss et al., 2021; Wheatley et al., 2007). This suggests our observed activations may extend beyond merely event or narrative processing. For marmosets, it's more challenging to make definitive conclusions as no previous fMRI studies have used the same animations. However, the new eye-tracking experiment results show marmosets spend more time focused on the red triangle in ToM videos but allocate similar minimal attention to the red triangle in both Random and GD videos. This suggests that marmosets process the Random and GD conditions similarly, but differently for ToM animations that represent mental interactions. Nevertheless, conclusive interpretation remains challenging, and this is indeed a matter that warrants further exploration. We appreciate the reviewer's critical perspective on this aspect of our study.

I wonder whether this interpretation might also reflect the curious finding of stronger medial PFC activation in Random trials versus ToM trials in humans, and no clear mPFC activation in the ToM trials. This seems very much at odds with the wider literature on the brain regions necessary for ToM, which often place the medial PFC at the heart of the social brain.

We appreciate the reviewer's observation concerning the surprising patterns of activation in the medial prefrontal cortex (mPFC) in our study. While many studies have indeed associated mPFC with ToM tasks, our findings of stronger mPFC activation during Random animations compared to ToM animations in humans, and the lack of clear mPFC activation in ToM trials, appear to diverge from the wider literature.

We would like to highlight that the role of mPFC in ToM may be more nuanced. For instance, a recent meta-analysis conducted by Schurz et al. (2021) demonstrated that social animation tasks, such as the Frith-Happé animated triangles task we used, tend to engage an intermediate cluster between cognitive and affective clusters. This cluster involves a variety of brain regions, including temporo-parietal areas, anterior temporal areas, dorso-posterior medial prefrontal cortex, and inferior frontal areas. This suggests that such tasks may not uniformly engage the entirety of the mPFC and might involve more the dorsal part.

Our study revealed stronger activations in the ventral part of the mPFC for Random versus ToM animations. These activations could reflect other processes, such as attentional control. Notably, our results align closely with those of the Human Connectome Project by Barch et al. (2013), who also observed stronger activations for Random versus ToM animations in similar ventral parts of the mPFC.

These observations underscore the complexity of the neural substrates of ToM and the potential influence of task designs on the patterns of brain activation. We concur that further research is needed to fully understand these complex issues, and we greatly appreciate the reviewer's contribution to this ongoing discussion.

While previous fMRI studies using the Frith-Happé task have found dorsal mPFC activation for the ToM animations, we did not observe a clear activation pattern in ToM trials in our study. This discrepancy could be attributable to a variety of factors. Even minor differences in task design or the specific versions of the animations used could lead to different cognitive processes being engaged during the task. The methodological aspects, such as statistical power, could also have contributed to the differences in our findings. Furthermore, the number of participants can affect the resulting brain activation patterns.

We added this explanation concerning the possible issues in the Discussion section on page 17, which now read:

“Our slightly adapted versions of the Frith-Happé animations led to a similar distinct pattern of brain activations, with an exception for the lack of activations in the dorsal part of the medial prefrontal cortex. This discrepancy could be attributable to various factors, including differences in task design, methodological aspects such as statistical power, or variations in participants characteristics.”

I very much appreciated the check using the independent HCP dataset. This was a very nice inclusion to ensure that the shortened version corresponded well with previous reports.

We are pleased to hear that the reviewer appreciates our use of the independent HCP dataset to validate our results. We thank the reviewer for the positive feedback on this aspect of our study.

Reviewer #2 (Recommendations for the authors):In their study, Dureux and colleagues are investigating the sensitivity of a highly social non-human primate species, the marmoset, to social illusion using the Frith and Happe task. Although this task is often considered a non-verbal TOM task, its relevance to investigate TOM has been disputed. For instance, the Frith and Happe task does not recruit in humans a similar network as other false-belief tasks and social gambling tasks (Schurz et al., 2020). While the authors might want to revise, or at least discuss their use of the TOM concept further, their results clearly show that marmosets distinguish the two types of videos shown to them.

We are grateful for the reviewer's thoughtful comments and critique. We appreciate the thoughtful reference to the study by Schurz et al. (2020). Indeed, as the reviewer has rightly highlighted, the Frith-Happé task has been debated for its relevancy to investigate ToM, given the divergence in neural networks it recruits compared to other ToM tasks, such as false-belief tasks and social gambling tasks, as demonstrated by Schurz et al. (2020).

In light of these insightful comments, we have revised our manuscript to acknowledge these differences and argue that the Frith-Happé task still holds value in the study of social cognition, albeit with a potential focus on different facets of this complex construct compared to more traditional ToM tasks. This understanding is reflected in our findings, showing that both humans and marmosets can distinguish between the types of videos in the Frith-Happé task.

The revised sections in our manuscript addressing these issues now read:

Page 17: “Overall, the ToM network we identified, as well as that reported by Barch et al. (2013), appear to be more extensive than those described in studies employing more complex experimental paradigms to study ToM. This aligns with the recent meta-analysis conducted by Schurz and colleagues (2020), which demonstrated that the network activated by simpler, non-verbal stimuli like social animations differs from the traditional network, with involvement of both cognitive and affective networks (Schurz et al., 2020).”

Page 20: “As such, future research including a range of tasks, from sensory-affective components to more abstract and decoupled representations of others' mental states (Schurz et al., 2020), will be fundamental in further unravelling the complexities of the evolution and functioning of the theory of mind across the primate lineage.”

We believe these revisions provide a more nuanced understanding of our study within the larger context of ToM research, and we thank the reviewer for prompting this important discussion. We address all the other points raised, below and in the manuscript.

Were there any differences between the different types of videos used? For instance, was there any difference between videos in which the interaction between the two shapes is more obvious (e.g. coaxing vs seducing).

We appreciate the reviewer's interest in the distinct types of videos used in our study. In this investigation, we did not specifically analyze the responses to videos where the interaction between the two shapes was more or less pronounced, such as in coaxing versus seducing scenarios. Our primary focus was on contrasting the overall responses elicited by Theory of Mind (ToM) animations and Random animations. Due to the design of our experiment, we did not include a sufficient number of repetitions for each distinct type of video within each condition to afford the statistical power necessary for such a comparison. We acknowledge that assessing responses to different types of ToM animations may provide additional insights into the specificity of neural responses and consider this an interesting avenue for future research.

Did the authors also present social videos to their animals? If so did they also observe additional recruitment of IPa and TPO areas like Clery and colleagues for social videos compared to Frith and Happe's videos (Cléry et al., 2021)?

In this study, our main focus was to examine the neural responses to the Frith-Happé animations, which represent abstract social scenarios. As such, we did not include realistic social videos to our animals in our current experimental design.

The study by Cléry et al. (2021) indeed demonstrates that the comparison social versus non-social realistic videos predominantly revealed a fronto-parietal network with additional temporal region engagement. The social condition in their study seems to recruit not only several areas that we observed to be activated in our ToM condition, but also additional temporal regions such as the IPa, TPO and TE1 areas. This suggests that these areas may play a significant role in the processing of more realistic social cues, thereby adding complexity to the social brain network.

Although our current study did not directly investigate this aspect, we agree that comparing the neural responses elicited by abstract versus realistic social scenarios in marmosets could provide valuable insights into the extent and adaptability of the social brain network. This approach would further our understanding of the neural substrates that underpin various facets of social cognition, depending on the complexity and realism of the presented stimuli. In future research, we plan to consider including such comparative investigations in our experimental design. We appreciate the reviewer's suggestion.

Humans and marmosets recruit a distinct set of subcortical structures during the viewing of video clips; for instance dorsal thalamus and cerebellum in humans, and the hippocampus and amygdala in marmosets. How do the authors interpret this difference?

We appreciate the reviewer for pointing out this difference in subcortical recruitment between humans and marmosets during the viewing of ToM versus Random video clips. The divergent patterns may reflect species-specific aspects of social cognition.

In humans, the involvement of the dorsal thalamus, which serves as a critical hub for information relay between various subcortical areas and the cortex, may indicate the necessity for complex information processing in interpreting the animations (e.g., Halassa and Sherman, 2019). The activation of the cerebellum, beyond its traditional role in motor functions, supports recent findings of its involvement in social cognition (e.g., Van Overwalle et al., 2014). The activations in a small portion of the amygdala may reflect emotional processing tied to understanding of the social scenarios in the animations (e.g., Janak and Tye, 2015).

On the other hand, the activation of the hippocampus and amygdala in marmosets might reflect a more emotion-driven interpretation of the animations. The recruitment of the hippocampus could suggest the role of memory in interpreting the animations by remembering past interactions to help interpret current social scenarios (e.g., Eichenbaum, 2017), while the more extended activation of the amygdala in marmosets might imply a higher degree of emotional processing compared to humans (e.g., Janak and Tye, 2015).

We have now expanded our discussion on this topic in the revised manuscript, specifically on pages 18-19:

"Regarding the distinct subcortical activations observed in humans and marmosets, it's important to consider the specific social cognitive demands that might be unique to each species. The involvement of the dorsal thalamus, cerebellum, and a small portion of the amygdala in humans may reflect the complexities of information processing, social cognition, and emotional involvement required to interpret the ToM animations (e.g., Halassa and Sherman, 2019; Janak and Tye, 2015; Van Overwalle et al., 2014). Conversely, the activation of the amygdala and hippocampus in marmosets could suggest a more emotion- and memory-based processing of the social stimuli (e.g., Eichenbaum, 2017; Van Overwalle et al., 2014). However, it's critical to consider that these interpretations are speculative and would require further study for confirmation.”

Unlike marmosets, rhesus macaques are not sensitive to the type of the Frith and Happe social illusion (Roumazeilles et al., 2021; Schafroth et al., 2021). The authors might want to discuss the singularity of the marmosets from an evolutionary perspective.

As suggested, we have now discussed the singularity of the marmosets from an evolutionary perspective in the Discussion section on pages 19-20, which reads:

“Interestingly, our results differed from those obtained by Roumazeilles et al. (2021) in their fMRI study conducted in macaques using the same animations. Roumazeilles and colleagues reported no differences in activation between ToM and Random animations, suggesting that rhesus macaques may not respond to the social cues presented by the ToM Frith-Happé animations. This disparity between our marmoset findings and those of macaques raises intriguing questions about potential differences in the evolutionary development of ToM processing within non-human primates. Marmosets, as New World monkeys, are part of an evolutionary lineage that diverged earlier than the lineage of Old-World monkeys such as macaques. This difference in lineage might lead to distinct evolutionary trajectories in cognitive processing, which could include varying sensitivity to abstract social cues in animations.

In summary, our study reveals novel insights into how New World marmosets, akin to humans, differentially process abstract animations that depict complex social interactions and animations that display purely physical or random movements. Our findings, supported by both specific gaze behaviors (i.e., the proportion of time spent on the red triangle, despite the inconclusiveness of overall fixation) and distinct neural activation patterns, shed light on the marmosets' capacity to interpret social cues embedded in these animations.

The differences observed between humans, marmosets, and macaques underscore the diverse cognitive strategies that primate species have evolved to decipher social information. This diversity may be influenced by unique evolutionary pressures that arise from varying social structures and lifestyles. Like macaque monkeys, humans often live in large, hierarchically organized social groups where status influences access to resources. However, both humans and marmosets share a common trait: a high degree of cooperative care for offspring within the group, with individuals other than the biological parents participating in child-rearing. These distinctive social dynamics of marmosets and humans may have driven the development of unique social cognitive abilities. This could explain their enhanced sensitivity to abstract social cues in the Frith-Happé animations.

Nonetheless, it is crucial to emphasize that even though marmosets respond to the social cues in the Frith-Happé animations, this does not automatically imply that they possess mental-state attributions comparable to humans. As such, future research including a range of tasks, from sensory-affective components to more abstract and decoupled representations of others' mental states (Schurz et al., 2020), will be fundamental in further unravelling the complexities of the evolution and functioning of the theory of mind across the primate lineage.”

Reviewer #3 (Recommendations for the authors):This study is strong in many ways, and the goal is a good one. The below recommendations will help strengthen it further:

We are truly appreciative of the reviewer's thorough assessment and constructive feedback on our manuscript. We have addressed all raised points both in our responses below and in the revised manuscript.

In the abstract, the claim that there is no evidence that nonhuman primates attribute mental states to moving shapes is false. You even cite some of this positive evidence (e.g., Uller, 2004; Atsumi et al., 2015; 2017). There is also evidence that they don't (Kupferberg et al., 2013; Burkart et al., 2012; Schafroth et al., 2021). The abstract would be stronger if written to represent the state of the field more accurately.

We thank the reviewer for drawing our attention to this. We've updated the abstract to reflect the current state of the field and our findings more accurately. Please refer to page 2 for the revised version.

Page 2: “Theory of Mind (ToM) refers to the cognitive ability to attribute mental states to other individuals. This ability extends even to the attribution of mental states to animations featuring simple geometric shapes, such as the Frith-Happé animations in which two triangles move either purposelessly (Random condition), exhibit purely physical movement (Goal-directed condition), or move as if one triangle is reacting to the other triangle’s mental states (ToM condition). While this capacity in humans has been thoroughly established, research on nonhuman primates has yielded inconsistent results.

This study explored how marmosets (Callithrix jacchus), a highly social primate species, process Frith-Happé animations by examining gaze patterns and brain activations of marmosets and humans as they observed these animations. We revealed that both marmosets and humans exhibited longer fixations on one of the triangles in ToM animations, compared to other conditions. However, we did not observe the same pattern of longer overall fixation duration on the ToM animations in marmosets as identified in humans. Furthermore, our findings reveal that both species activated extensive and comparable brain networks when viewing ToM versus Random animations, suggesting that marmosets differentiate between these scenarios similarly to humans. While marmosets did not mimic human overall fixation patterns, their gaze behavior and neural activations indicate a distinction between ToM and non-ToM scenarios. This study expands our understanding of nonhuman primate cognitive abilities, shedding light on potential similarities and differences in ToM processing between marmosets and humans.”

The overall conclusion as stated in the abstract, at the end of the introduction, and in the discussion is not warranted by the evidence. Indeed, the abstract completely fails to mention that the marmosets failed to show the human-like pattern of longer fixations on the ToM videos. Many readers will likely interpret this evidence as primarily against the idea that marmosets view the ToM videos in a human-like way, or as equivocal evidence at best. This report will be a stronger piece of science if it accurately describes the results.

We agree with the reviewer's comment regarding the absence of certain result descriptions in the abstract, introduction, and discussion. In response to this feedback, we have conducted major revisions in these three sections, adding the missing information and elaborating on the overall conclusion. The primary changes can be found on page 2 (abstract), pages 4-5 (introduction), and pages 14 to 20 (Discussion). The revised content reads as follows:

Page 2: “Theory of Mind (ToM) refers to the cognitive ability to attribute mental states to other individuals. This ability extends even to the attribution of mental states to animations featuring simple geometric shapes, such as the Frith-Happé animations in which two triangles move either purposelessly (Random condition), exhibit purely physical movement (Goal-directed condition), or move as if one triangle is reacting to the other triangle’s mental states (ToM condition). While this capacity in humans has been thoroughly established, research on nonhuman primates has yielded inconsistent results.

This study explored how marmosets (Callithrix jacchus), a highly social primate species, process Frith-Happé animations by examining gaze patterns and brain activations of marmosets and humans as they observed these animations. We revealed that both marmosets and humans exhibited longer fixations on one of the triangles in ToM animations, compared to other conditions. However, we did not observe the same pattern of longer overall fixation duration on the ToM animations in marmosets as identified in humans. Furthermore, our findings reveal that both species activated extensive and comparable brain networks when viewing ToM versus Random animations, suggesting that marmosets differentiate between these scenarios similarly to humans. While marmosets did not mimic human overall fixation patterns, their gaze behavior and neural activations indicate a distinction between ToM and non-ToM scenarios. This study expands our understanding of nonhuman primate cognitive abilities, shedding light on potential similarities and differences in ToM processing between marmosets and humans.”

Pages 4-5: “Although the spontaneous attribution of mental states to moving shapes has been well established in humans, it remains uncertain whether other primate species share this capacity. There is some evidence suggesting that monkeys can attribute goals to agents with varying levels of similarity and familiarity to conspecifics, including human agents, monkey robots, moving geometric boxes, animated shapes, and simple moving dots (Atsumi et al., 2017; Atsumi and Nagasaka, 2015; Krupenye and Hare, 2018; Kupferberg et al., 2013; Uller, 2004). However, the findings in this area are somewhat mixed, with some studies investigating the attribution of goals to inanimate moving objects yielding inconclusive results (Atsumi and Nagasaka, 2015; Kupferberg et al., 2013). Nonhuman primates' spontaneous attribution of mental states to Frith-Happé animations is even less certain. While human subjects exhibit longer eye fixations when viewing the ToM condition compared to the Random condition of the Frith-Happé animations (Klein et al., 2009), a recent eye tracking study in macaque monkeys did not observe similar differences (Schafroth et al., 2021). Similarly, a recent fMRI study conducted on macaques found no discernible differences in activations between ToM and random Frith-Happé animations (Roumazeilles et al., 2021).

In this study, we investigated the behaviour and brain activations of New World common marmoset monkeys (Callithrix jacchus) while they viewed Frith-Happé animations. Living in closely-knit family groups, marmosets exhibit significant social parallels with humans, including prosocial behavior, imitation, and cooperative breeding. These characteristics establish them as a promising nonhuman primate model for investigating social cognition (Burkart et al., 2009; Burkart and Finkenwirth, 2015; Miller et al., 2016). To directly compare humans and marmosets in their response to these animations, we employed high-speed video eye-tracking to record eye movements in eleven healthy humans and eleven marmoset monkeys. Additionally, we conducted ultra-high field fMRI scans on ten healthy humans at 7T and six common marmoset monkeys at 9.4T. These combined methods allowed us to examine the visual behavior and brain activations of both species while they observed the Frith-Happé animations.”

Pages 14-15: “In our first experiment, we examined the gaze patterns of marmosets and humans during the viewing of these video animations. Klein et al. (2009) reported differing fixation durations for these animations, where the longest fixations were observed for ToM animations, followed by GD animations and the shortest fixations for Random animations. They further reported that the intentionality score – derived from verbal descriptions of the animations – followed a similar pattern: highest for ToM, lowest for Random, and intermediate for GD animations. This validated the degree of mental state attribution according to the categories and established that animations provoking mentalizing (ToM condition) were associated with long fixations. This, in turn, supports the use of fixation durations as a nonverbal metric for mentalizing capacity (Klein et al., 2009; Meijering et al., 2012). Our results with human subjects, which demonstrated longer fixation durations for the ToM animations compared to the GD and Random animations, paralleled those of Klein et al. (2009). However, unlike Klein et al.'s findings, we did not observe intermediate durations for GD animations in our study.

Interestingly, our marmoset data did not align with the human findings but instead resonated more with Schafroth et al. (2021)'s observations in macaque monkeys, which did not show significant differences in fixation durations across the three animation types.”

Pages 19-20: “In summary, our study reveals novel insights into how New World marmosets, akin to humans, differentially process abstract animations that depict complex social interactions and animations that display purely physical or random movements. Our findings, supported by both specific gaze behaviors (i.e., the proportion of time spent on the red triangle, despite the inconclusiveness of overall fixation) and distinct neural activation patterns, shed light on the marmosets' capacity to interpret social cues embedded in these animations.

The differences observed between humans, marmosets, and macaques underscore the diverse cognitive strategies that primate species have evolved to decipher social information. This diversity may be influenced by unique evolutionary pressures that arise from varying social structures and lifestyles. Like macaque monkeys, humans often live in large, hierarchically organized social groups where status influences access to resources. However, both humans and marmosets share a common trait: a high degree of cooperative care for offspring within the group, with individuals other than the biological parents participating in child-rearing. These distinctive social dynamics of marmosets and humans may have driven the development of unique social cognitive abilities. This could explain their enhanced sensitivity to abstract social cues in the Frith-Happé animations.

Nonetheless, it is crucial to emphasize that even though marmosets respond to the social cues in the Frith-Happé animations, this does not automatically imply that they possess mental-state attributions comparable to humans. As such, future research including a range of tasks, from sensory-affective components to more abstract and decoupled representations of others' mental states (Schurz et al., 2020), will be fundamental in further unravelling the complexities of the evolution and functioning of the theory of mind across the primate lineage.”

The justification for looking in marmosets could be read to imply that macaque monkeys do not live in family groups or share important social similarities with humans. Both species share many social similarities (and many social differences) with humans. Marmosets are a good species to study; this section would benefit from a more accurate rationale.

We apologize for any confusion our previous wording may have caused. We have now revised the sentence on page 4 to enhance its clarity and accuracy, which now reads:

Page 4: “Living in closely-knit family groups, marmosets exhibit significant social parallels with humans, including prosocial behavior, imitation, and cooperative breeding. These characteristics establish them as a promising nonhuman primate model for investigating social cognition (Burkart et al., 2009; Burkart and Finkenwirth, 2015; Miller et al., 2016).”

Because it is one of the main metrics in the Klein and Schafroth papers, and thus readers will want to see it for sake of comparison, the authors should include a figure showing the overall fixation durations as a function of category and species.

In response to the reviewer’s suggestion, we have added a new figure (Figure 2) on page 39, which presents the overall fixation durations as a function of both animation category and species. This figure provides a direct comparison of fixation durations between humans and marmosets across the different animation conditions (Random, Goal-directed, and ToM). We believe this additional visualization will assist readers in better understanding the overall durations of fixation across species and conditions and enable direct comparison with the findings of Klein and Schafroth.

The results about looking time to the large triangle need to follow up on the interaction between species and conditions so that readers know how to interpret it.

As we have now included the Goal-directed condition in our eye-tracking experiment, we have substantially revised the section “Gaze patterns for Frith-Happé’s ToM, GD and Random animations in humans and marmosets” in the results (pages 5 to 7). In response to the reviewer's comments, we have provided a more detailed analysis of the interaction between species and conditions regarding the looking time spent on the large triangle, which now reads as follows:

Pages 6-7: “To further analyze the gaze patterns of both humans and marmosets, we next measured the proportion of time subjects looked at each of the triangles in the videos (Figure 2B). We conducted mixed ANOVAs on the proportion of time the radial distance between the current gaze position and each triangle was within 4 visual degrees for each triangle separately.

Importantly, we observed a significant interaction between species and condition for the proportion of time spent looking at the large red triangle (F(2,40)=9.83, p<.001, ηp2 = .330). Specifically, both humans (Figure 2B left) and marmosets (Figure 2B right) spent a greater proportion of time looking at the red triangle in ToM compared to the GD and Random videos (For humans, ToM vs GD: Δ=.23, p<.001 and ToM vs Random: Δ=.31, p<.001 ; For marmosets, ToM vs GD: Δ=.13, p<.01 and ToM vs Random: Δ=.13, p<.01). However, while humans also allocated a greater proportion of time to the red triangle in GD compared to Random animations (Δ=.08, p=.05), marmosets did not show any difference between these two conditions (Δ=.0003, p=1).

For the small blue triangle, we also observed a significant interaction of species and condition (F(2,40)=3.54, p=.04, ηp2=.151) but the comparisons were not resistant to the p value adjustment by Bonferroni correction. Therefore, humans and marmosets spent the same proportion of time looking at the blue triangle in the three different types of videos (For humans, ToM vs GD: Δ=-.02, p=1, ToM vs Random: Δ=.04, p=1 and GD vs Random: Δ=.07, p=.23 ; For marmosets, ToM vs GD: Δ=-.05, p=.89, ToM vs Random: Δ=.07, p=.66 and GD vs Random: Δ=-.02, p=1; Figure 2B).

These results highlight the variation in gaze patterns observed in both humans and marmosets when their focus is directed towards the large red triangle during the viewing of ToM, GD, and Random videos. Notably, humans show a gradient of proportion of time spent looking at the red triangle across the three conditions, with the smallest proportion in Random videos and the greatest proportion in ToM videos. In contrast, marmosets exhibit a different pattern, spending more time looking at the red triangle in ToM videos, but allocating the same proportion of time to look at the red triangle in both Random and GD videos. This finding suggests that while humans demonstrate distinct attentional preferences for the red triangle across the three conditions, marmosets exhibit a similar attentional focus on the red triangle in the Random and GD conditions, but their pattern differs in the ToM condition. This suggests that marmosets process the Random and GD conditions in a similar manner, but their processing of the ToM condition is distinct, indicating a differential response to stimuli representing social interactions.”

The sentence on lines 97-99 might be an incomplete sentence.

We appreciate the reviewer's attention to detail and acknowledge the oversight in the sentence structure on lines 97-99. We have revised this sentence and the entire paragraph on pages 5 to 7, under the heading "Gaze patterns for Frith-Happé’s ToM, GD and Random animations in humans and marmosets". This revision takes into account the new results obtained after adding the Goal-Directed condition to the experiment.

Are the bars in Figure 2 meant to add up to 1 for any given participant? If you analyzed the total time fixating on either shape, would marmosets be spending less time looking at the shapes overall than humans?

We thank the reviewer for the question. The values in the previous Figure 2 are not intended to add up to 1. This is because there are instances where the triangles overlap or are in close enough proximity that the eye position falls within the defined radius for both simultaneously. In response to the second question, we have conducted an analysis on the total time spent fixating on either shape. Our findings revealed a significant effect of species (*F*_(1,20)_=14.38, *p*=.001, *η_p_^2^*=.42), indicating that humans tend to look at the triangles more frequently than marmosets (Δ=.16, *p*=.001).

Overall, the figures are quite informative and aesthetically pleasing.

We sincerely thank the reviewer for their positive remarks about the figures in our study.

HCP should be explained the first time it is used.

We agree with the reviewer's point that all abbreviations should be clearly explained when first introduced. We have now amended the text to clarify this at the first instance where HCP appears on page 10. We thank the reviewer for bringing this oversight to our attention.

Readers will likely want clarification in cases where the same area showed stronger activation for ToM videos AND Random videos. I assume it was in different voxels in the same larger area, but this could be explicit.

We appreciate the reviewer's suggestion and agree that clarity on this issue is essential.

We have now added a sentence on this point in the two relevant sections of fMRI results on humans and marmosets, on pages 7-8 and 11, to ensure this is explicitly stated and clear to the reader. This can be read:

Pages 7-8 (Functional brain activations while watching ToM and Random Frith-Happé’s animations in humans): “Both ToM (Figure 3A) and Random (Figure 3B) videos activated a large bilateral network. While the same larger areas were activated in both conditions, the specific voxels showing this activation within those areas were typically distinct. In some cases, both conditions activated the same voxels, but the degree of activation differed. This suggests a degree of both spatial and intensity variation in the activations for the two conditions within the same areas. (…)”

Page 11 (Functional brain activations while watching ToM and Random Frith-Happé’s animations in marmosets): “Both the ToM (Figure 4A) and Random (Figure 4B) animations activated an extensive network involving a variety of areas in the occipito-temporal, parietal and frontal regions. As in human subjects, it should be noted that while both conditions elicited strong activation in some of the same larger areas, these activations might have either occurred in distinct voxels within those areas, or the same voxels were activated to varying degrees for both conditions. This suggests distinct yet overlapping patterns of neural processing for the ToM and Random conditions.”

The claim that these maps represent "dedicated brain networks" for ToM or Random videos (line 188) is too strong. These brain areas are used for many things.

We agree with the reviewer's concern regarding the term "dedicated brain networks". We understand that the use of this term could be misinterpreted as implying exclusivity, which is not the case. These areas are indeed involved in various cognitive functions. We have modified the statement on page 10 to indicate that these are "brain networks activated during the processing of ToM or Random videos" instead of "dedicated brain networks". We appreciate this valuable input.

For many of the sentences in the imaging results, the comparison needs to be made explicit. For example Line 193 – higher bilateral activation than what? Line 196 – greater activations than what? Line 202 – a larger network than what? Etc.

We appreciate the reviewer's attention to detail in pointing out the need for clear and explicit comparisons in our imaging results. We recognize that some sentences may lack specificity, leading to potential confusion. We have now revised these sentences in the Results section to clearly specify the comparisons being made in each case. The updated descriptions can be found in the Results section, on pages 7 to 12.

The description of Klein et al., (2009) on Lines 289-293 might be read to imply that they were attributing mentalizing without good reason. Klein also collected intentionality scores, which correlated with the viewing metric. This could be rephrased to be more accurate.

Thank you for pointing out the potential misinterpretation of our description of Klein et al., 2009. Our intention was not to undermine the work by Klein et al. We have revised the phrasing in our manuscript, on page 15, to better reflect this aspect of their study:

Page 15: “In our first experiment, we examined the gaze patterns of marmosets and humans during the viewing of these video animations. Klein et al. (2009) reported differing fixation durations for these animations, where the longest fixations were observed for ToM animations, followed by GD animations and the shortest fixations for Random animations. They further reported that the intentionality score – derived from verbal descriptions of the animations – followed a similar pattern: highest for ToM, lowest for Random, and intermediate for GD animations. This validated the degree of mental state attribution according to the categories and established that animations provoking mentalizing (ToM condition) were associated with long fixations. This, in turn, supports the use of fixation durations as a nonverbal metric for mentalizing capacity (Klein et al., 2009; Meijering et al., 2012).”

In general, the discussion could be strengthened by avoiding repeating the results in as much detail.

We appreciate the reviewer's feedback and agree that a more concise discussion could make the manuscript more effective. We have revised the Discussion section to provide a more focused analysis and interpretation of the results, limiting repetition from the Results section.

The updated Discussion section can be found from page 14 to 20 of the revised manuscript.

The inclusion of the authors as subjects is odd. Some readers will view it as a big red flag. The authors clearly know their own hypothesis and likely have a vested interest in a particular outcome. For the strongest report, the authors should remove their own data. At the very least, the authors need to demonstrate that the inclusion/exclusion of their unblinded data doesn't affect the interpretation of the human results.

We appreciate the reviewer's concern about the potential bias introduced by the inclusion of authors as subjects in our study. Taking this into consideration, we have removed the data from the three authors who initially participated in the study. For the eye tracking experiment, we have now introduced a new “Goal-Directed condition” and conducted the experiment with eleven new subjects, none of whom are authors of this study. For the fMRI experiment, we replaced the data from the three author-subjects with data from three additional subjects who were not privy to the study's hypothesis.

Consequently, we replaced the previous results with these new findings and made the necessary modifications to several sections of the manuscript as well as on the figures 2, 3 and 5 and the figures supplement 1 and 2. These updates can be found in the sections: “Gaze patterns for Frith-Happé’s ToM, GD and Random animations in humans and marmosets” on pages 5 to 7, “Functional brain activations while watching ToM and Random Frith-Happé’s animations in humans” on pages 7 to 10, and “Comparison of functional brain activations in humans and marmosets” on pages 13-14. The updated figures can be found on pages 39, 40, 42, 43, and 44, respectively.

We have also updated the participant information in the methods section, on page 22, to now read:

“Eleven healthy humans (4 females, 25-42 years, mean age: 30.7 years) participated in the eye tracking experiment. Among these, five individuals, along with eight additional subjects (4 females, 26-45 years), took part in the fMRI experiment.”

The method should state whether the subjects had experienced these animations before (e.g., they're shown in some psychology and neuroscience classes).

Thank you for pointing out the necessity to include this information. We understand that the participants' previous exposure to these animations could potentially affect the results. We have added the following sentence to the Methods section:

Page 22: “Importantly, all subjects confirmed they had not previously been exposed to the Frith-Happé animation videos used in our study.”

If the authors proceed in pushing this data without the Goal-Directed videos, they need to at least address their rationale for not testing these videos.

We appreciate this important point, raised also by the first reviewer. As previously mentioned, our initial strategy was to focus on the two extreme conditions: ToM and Random, representing scenarios with and without mental interactions, respectively. This choice was influenced by some previous fMRI studies using the Frith-Happé animated triangles task in humans and macaques, which predominantly examined these two conditions (Gobbini et al., 2007; Barch et al., 2013; Bliksted et al., 2019; Vandewouw et al., 2021; Weiss et al., 2021; Chen et al., 2023; Roumazeilles et al., 2021). We also faced practical challenges in incorporating all conditions with a sufficient number of repetitions into the fMRI task design for marmoset subjects, given the substantial duration of each video clip (19.5 sec). As such, we chose to focus our investigation on the ToM and Random conditions, as the Goal-Directed (GD) condition falls between these two extremes, depicting physical interaction among the triangles without suggesting mental state attribution. Recognizing the potential limitations of not including the GD condition, we conducted an additional eye-tracking experiment that encompassed all three conditions.

We have updated the manuscript to clarify the choice of conditions for the fMRI experiment and to incorporate the findings from the new eye-tracking experiment. Relevant modifications have been made in the methods, results, and Discussion sections. These revisions can be found on pages 23-24, 5 to 7, and 14 to 20, respectively.

The description of the monkey reward contingencies needs to be clearer about whether the monkeys were rewarded only during calibration or during videos as well, and whether any reward during videos was contingent on keeping their eyes on the screen.

We apologize for any previous ambiguity in the text. It is important to clarify that the monkeys were rewarded solely during the initial and final stages of the sessions, and no rewards were administered during the calibration or the experiment. Accordingly, we have updated the description in the Methods section, now stated on page 23 as:

“Monkeys were rewarded at the beginning and end of each session. Crucially, no rewards were provided during the calibration or while the videos were played.”

Because this is a social task when the scans were normalized to MNI space, did the authors divide the human participants into those with and without a paracingulate sulcus?

We thank the reviewer for the insightful question. Indeed, in this study, we normalized the MRI scans of human participants to MNI space, providing a standardized representation of the brain. However, we did not separate participants based on the presence or absence of a paracingulate sulcus in our analysis. Your suggestion to incorporate this anatomical variability is intriguing, especially given its potential implications in social cognition research. We appreciate this thoughtful suggestion and will certainly consider it in our future studies.

The authors need to better specify what counts as a "baseline" for the fMRI comparisons. They should also briefly justify why this is an informative comparison.

Apologies for any confusion regarding our baseline condition in the fMRI comparisons. In this study, our baseline refers to the brain's activity when the subjects are not engaged in the tasks (i.e., viewing ToM or Random animations). More specifically, we have defined the baseline as the brain activity during the presentation of a circular black cue between video clips. Selecting this as the baseline is crucial as it presents a 'resting state' scenario – a state where the brain is not actively engaged in processing task-specific stimuli but is instead in a neutral, alert state. This choice of baseline allows us to identify and compare increased activity in different brain regions during the ToM and Random conditions relative to this resting state. This, in turn, aids our understanding of the specific functional brain regions associated with the processing of these specific conditions. We have now clarified this point in the Methods section of the manuscript on page 30. It now reads:

“First, we identified brain regions involved in the processing of ToM and Random animations by contrasting each condition with a baseline (i.e., ToM condition > baseline and Random condition > baseline contrasts). This baseline brain activation recorded during the presentation of the circular black cue between video clips (i.e., baseline blocks of 15 sec, see above), reflects 'resting state' activation. By comparing it to the brain activation during ToM and Random animations, we could specifically highlight the task-related activations and isolate brain regions engaged during each condition.”